# Sharpness of Minima in Deep Matrix Factorization: Exact Expressions

## Abstract

Understanding the geometry of the loss landscape near a minimum is key to explaining the implicit bias of gradient-based methods in non-convex optimization problems such as deep neural network training and deep matrix factorization. A central quantity to characterize this geometry is the maximum eigenvalue of the Hessian of the loss, which measures the sharpness of the landscape. Currently, its precise role has been obfuscated because no exact expressions for this sharpness measure were known in general settings. In this paper, we present the first exact expression for the maximum eigenvalue of the Hessian of the squared-error loss at *any* minimizer in general overparameterized deep matrix factorization (i.e., deep linear neural network training) problems, resolving an open question posed by Mulayoff & Michaeli (2020). This expression uncovers a fundamental property of the loss landscape of depth-2 matrix factorization problems: *a minimum is flat if and only if it is spectral-norm balanced*, which implies that *flat minima are not necessarily Frobenius-norm balanced*. Furthermore, to complement our theory, we empirically investigate an escape phenomenon observed during gradient-based training near a minimum that crucially relies on our exact expression of the sharpness.

## 1 Introduction

Decades of research in learning theory suggest limiting model complexity to prevent overfitting. However, modern deep learning is heavily overparameterized and has nonetheless achieved unprecedented success in practice over the past decade (Krizhevsky et al., 2012; Vaswani et al., 2017). Generally, in overparameterized settings, the loss function has infinitely many global minima that achieve zero training error (interpolation regime), yet these models still perform well. This phenomenon has been explored in various settings such as nonparametric regression, (Belkin et al., 2019), training two-layer neural networks with logistic loss (Frei et al., 2022), and linear regression (Bartlett et al., 2020).

The propensity of neural network training dynamics to converge to *good minima* is attributed to the ability of gradient-based optimization algorithms to avoid *bad minima* (Neyshabur et al., 2017; Zhang et al., 2017). This is related to the *implicit bias* of gradient descent (GD) (Neyshabur et al., 2014), and a large body of work has focused on its understanding (Gunasekar et al., 2017; 2018; Soudry et al., 2018; Arora et al., 2019; Ji & Telgarsky, 2020; Yun et al., 2021).

It has been observed that *dynamical stability* of GD near a minimum is a key factor in characterizing its implicit bias toward particular solutions (Wu et al., 2018; Nar & Sastry, 2018). Conceptually, dynamical stability refers to the ability of GD to *stably converge* to a minimum, and it is closely related to the sharpness of the loss landscape in its vicinity (Mulayoff et al., 2021). This topic has been investigated in numerous works (Nar & Sastry, 2018; Wu et al., 2018; Ma & Ying, 2021; Mulayoff et al., 2021; Nacson et al., 2023; Qiao et al., 2024; Liang et al., 2025) within the framework of the classical notion of *linear stability* in dynamical systems (Strogatz, 2024).

Ultimately, this understanding boils down to understanding the *geometry* of the loss landscape near a minimum. The maximum eigenvalue of the Hessian of the loss serves as a key measure to quantify the *sharpness* of the landscape near a minimum. Despite its significance, its precise role is not well-understood, particularly because closed-form expressions are generally unknown, outside a

Table 1: Closed-form expressions for the maximum Hessian eigenvalue in the literature. $\Omega$ denotes the set of *all* global minimizers, $\Omega_F \subseteq \Omega$ denotes the set of *flat* global minimizers, and $\Omega_B \subseteq \Omega$ denotes the set of *balanced* global minimizers.

| Related Work | Depth | Input Dim. | Output Dim. | $\lambda_{\max}(\nabla^2 \mathcal{L}(\boldsymbol{w}))$ | Layers |
|---|---|---|---|---|---|
| Mulayoff & Michaeli (2020, Theorem 1) | $L$ | $d_0$ | $d_L$ | $\boldsymbol{w} \in \Omega_F$ | $\mathbb{R}^{a \times b}$ |
| Zhu et al. (2023, Appendix B.1) | 2 | 1 | 1 | $\boldsymbol{w} \in \mathbb{R}^N$ | $\mathbb{R}$ |
| Singh & Hofmann (2024, Theorem 1) | 2 | 1 | 1 | $\boldsymbol{w} \in \mathbb{R}^N$ | $\mathbb{R}^a$ |
| Ghosh et al. (2025, Lemma 1) | $L$ | $d_0$ | $d_L$ | $\boldsymbol{w} \in \Omega_B$ | $\mathbb{R}^{a \times b}$ |
| Theorem 5 (This Paper) | $L$ | $d_0$ | $d_L$ | $\boldsymbol{w} \in \Omega$ | $\mathbb{R}^{a \times b}$ |

few particular cases. We summarize the current state of understanding as well as the contributions of our paper in Table 1.

Most notably, the seminal work of Mulayoff & Michaeli (2020) derives a closed-form expression for the maximum eigenvalue of the Hessian at *flat* global minima of deep linear networks (i.e., deep matrix factorization) with squared-error loss. However, obtaining a closed-form expression for *all* global minima in deep linear networks/deep matrix factorization was an open problem. In particular, Mulayoff & Michaeli (2020) claim that finding a closed-form expression for arbitrary global minima is intractable. In this paper, we refute this claim and positively answer the following fundamental question.

*Does a closed-form expression for the maximum eigenvalue of the Hessian exist for overparameterized deep matrix factorization problems?*

In particular, in Theorem 5, we provide a closed-form expression for the maximum Hessian eigenvalue at arbitrary minima of depth-$L$ overparameterized deep matrix factorization. We also highlight that overparameterized deep matrix factorization and the deep linear neural network setting of Mulayoff & Michaeli (2020) are equivalent for investigating the sharpness measure $\lambda_{\max}(\nabla^2 \mathcal{L}(\boldsymbol{w}^*))$ when the data covariance matrix is identity, i.e., $\boldsymbol{\Sigma}_x = \boldsymbol{I}$ (see Mulayoff & Michaeli, 2020, Equation 21). To the best of our knowledge, our analysis provides the first exact expression of the maximum eigenvalue for deep matrix factorization/deep linear neural network problems. In the case of deep overparameterized scalar factorization (Theorem 4) and depth-2 matrix factorization (Corollary 6), our closed-form expression simplifies considerably.

With our closed-form expression in hand, we then empirically explore in Section 7 the *escape phenomenon*, observed by Wu et al. (2018) (who only studied a one-dimensional setting). We find that this phenomenon also occurs for overparameterized deep matrix factorization problems. Therefore, we empirically observe the following.

*GD escapes from a dynamically unstable minimum almost surely.*

We explore this phenomenon through the lens of *dynamical stability* introduced by Wu et al. (2018).

Consider a twice continuously differentiable loss function $\mathcal{L} : \mathbb{R}^N \to \mathbb{R}$, and let $\boldsymbol{x}^*$ be a minimizer of $\mathcal{L}$. For $\delta > 0$, denote the open ball $B_\delta(\boldsymbol{x}^*) := \left\{ \boldsymbol{x} \in \mathbb{R}^N : \|\boldsymbol{x} - \boldsymbol{x}^*\|_2 < \delta \right\}$. There exists a $\delta > 0$ such that, for all $\boldsymbol{x} \in B_\delta(\boldsymbol{x}^*)$,

$$\mathcal{L}(\boldsymbol{x}) = \mathcal{L}(\boldsymbol{x}^*) + (\boldsymbol{x} - \boldsymbol{x}^*)^\top \nabla \mathcal{L}(\boldsymbol{x}^*) + \frac{1}{2}(\boldsymbol{x} - \boldsymbol{x}^*)^\top \nabla^2 \mathcal{L}(\boldsymbol{x}^*)(\boldsymbol{x} - \boldsymbol{x}^*) + o(\|\boldsymbol{x} - \boldsymbol{x}^*\|_2^2). \quad (1)$$

**Definition 1.** *Let $\mathcal{L} : \mathbb{R}^N \to \mathbb{R}$ be a twice continuously differentiable loss function, and let $\boldsymbol{x}^*$ be a minimizer of $\mathcal{L}$. The corresponding linearized GD update rule for $\mathcal{L}$ is given by*

$$\boldsymbol{x}_{t+1} = \boldsymbol{x}_t - \eta \nabla^2 \mathcal{L}(\boldsymbol{x}^*)(\boldsymbol{x}_t - \boldsymbol{x}^*), \quad (2)$$

*where $\eta > 0$ is the step size. Given $\delta > 0$ such that (1) holds, if there exists an initial point $\boldsymbol{x}_0 \in B_\delta(\boldsymbol{x}^*)$ such that the residuals $\boldsymbol{\epsilon}_t = \boldsymbol{x}_t - \boldsymbol{x}^*$ diverge, i.e.,*

$$\lim_{t \to \infty} \|\boldsymbol{\epsilon}_t\|_2 = \infty, \quad (3)$$

*then $\boldsymbol{x}^*$ is said to be* dynamically unstable.

A necessary and sufficient condition for a minimum to be dynamically unstable is that $\lambda_{\max}(\nabla^2 \mathcal{L}(\boldsymbol{x}^*)) > 2/\eta$ (cf. Wu et al. (2018); Mulayoff et al. (2021); Chemnitz & Engel (2025)). Thus, we see that an exact expression for the maximum eigenvalue is necessary to explore the escape phenomenon empirically.

## 1.1 Contributions

In this paper, we present the first exact expression for the maximum eigenvalue of the Hessian of the squared-error loss at any minimizer in general overparameterized deep matrix factorization problems. Our results lead to the following remarkable observations:

- *A minimizer of deep overparameterized scalar factorization loss is flat **if and only if** the product of spectral norms of left and right intermediate factors is constant across layers (Corollary 7).*

- *Flat minima are spectral-norm balanced in depth-2 matrix factorization (Corollary 8). This implies that **flat minima are not necessarily balanced**, contrary to claims made in several works (Ding et al., 2024; Ghosh et al., 2025). We further discuss this in Section 6.*

- *A minimizer of deep matrix factorization loss is flat **if** the product of spectral norms of left and right intermediate factors is constant across layers (Corollary 10).*

- A recent work (Anonymous, 2025) provides empirical evidence that, in two-layer linear networks, the largest eigenvalue of the Hessian of the loss function is correlated with the spectral norm of the weight matrices. Furthermore, for deep linear networks, they observe that this largest Hessian eigenvalue correlates with the spectral norm of the products of the weight matrices. Our results in Theorem 4, Theorem 5, and Corollary 6 provide the first theoretical explanation for their empirical observations.

## 1.2 Related Work

**Flat Minima.** Mulayoff & Michaeli (2020) derived a closed-form expression for the maximum eigenvalue of the Hessian at flat minima for deep linear neural networks. They also showed that the Hessian at a global minimum is rank-deficient by at least the order of $1 - 1/L$, where $L$ is the depth of the network. Moreover, they showed that the sharpness of the flattest minima increases approximately linearly with $L$ if $L \gg 1$. Singh & Hofmann (2024) provided a full characterization of the Hessian spectrum at a point in parameter space for linear and ReLU networks in the *scalar regression* case. They observed that the eigenvalues scale in proportion to the input variance within one hidden-layer scalar linear networks. More recently, Josz (2025) has shown that locally flat minima are globally flat in depth-2 matrix factorization problems.

**Balanced Minima.** Ghosh et al. (2025) provided a full characterization of the Hessian spectrum at balanced minima in deep matrix factorization. Furthermore, they showed that the maximum eigenvalue of the Hessian at the flattest minima is equal to that of the balanced minima. Ding et al. (2024) showed that *norm-minimal*, *balanced*, and *flat* solutions coincide in depth-2 matrix factorization, where sharpness measured by the *scaled trace* of the Hessian matrix of the loss function. We further discuss Ghosh et al. (2025) and Ding et al. (2024)'s results in Section 6. Finally, Wang et al. (2022) showed that large step size GD training induces a *balancing effect* between factors in depth-2 matrix factorization.

**Dynamical Stability.** In dynamical systems theory, it is well established that asymptotic convergence to a critical point is determined solely by the local stability of that point (Strogatz, 2024). In the seminal work of Wu et al. (2018) on the dynamical stability analysis of GD training, it was shown that a global minimum is *dynamically stable* for GD if and only if the step size does not exceed $2/\lambda_{\max}$, where $\lambda_{\max}$ denotes the maximum eigenvalue of the Hessian of the loss at the minimum. Mulayoff et al. (2021) investigated this mechanism in the space of learned functions for two-layer overparameterized univariate ReLU networks in the interpolation regime. This was then extended to multivariate ReLU networks by Nacson et al. (2023). The interpolation assumption was then removed by Qiao et al. (2024); Liang et al. (2025).

**Edge-of-Stability.** Cohen et al. (2021) observed that neural networks trained with GD typically operate in a regime called *edge of stability*, in which the maximum eigenvalue of the Hessian of

the loss function hovers just above the value $2/\eta$, where $\eta$ is the step size, and argued that classical optimization theory fails to explain this phenomenon. Recently, Liang et al. (2025) empirically observed that explicit regularization seems to break the edge-of-stability phenomenon.

**Sharpness and Generalization.** It is widely recognized in the literature that flat minima are associated with better generalization (Hochreiter & Schmidhuber, 1997; Keskar et al., 2017). In a large-scale empirical investigation, Jiang et al. (2020) examined different measures for deep networks and found that a sharpness-based measure exhibited the strongest correlation with generalization. There is also theoretical evidence for this phenomenon in low-rank matrix recovery (Ding et al., 2024). On the other hand, Dinh et al. (2017) showed that *good minima* can be arbitrarily sharp in deep neural networks.

## 2 Notation, Preliminaries, and Problem Setup

We denote the *Kronecker product* by $\otimes$, the *Frobenius inner product* by $\langle \cdot, \cdot \rangle$, the *spectral norm* by $\sigma_{\max}(\cdot)$, and the *Frobenius norm* by $\|\cdot\|_F$. We denote by $[L]$ the set of natural numbers up to $L$, i.e., $[L] = \{1, 2, \ldots, L\}$.

To simplify the notation for subsequent derivations, we define

$$\prod_{j=n}^{m} \boldsymbol{W}_j := \begin{cases} \boldsymbol{W}_m \boldsymbol{W}_{m-1} \ldots \boldsymbol{W}_n & \text{if } n \leq m, \\ \boldsymbol{I}_{d_m} & \text{otherwise, where } n, m \in [L], \end{cases} \tag{4}$$

where $\boldsymbol{W}_m \in \mathbb{R}^{d_m \times d_{m-1}}$.

Our analysis relies on matrix calculus and the formulation of directional second derivatives. Therefore, before proceeding to the technical details, we find it useful to first develop the intuition behind directional derivatives of real-valued functions of matrix variables.

**Gâteaux Derivatives**. Let $f : \mathbb{R}^{K \times L} \to \mathbb{R}$ be a differentiable function with continuous first- and second-order derivatives on $\mathbb{R}^{K \times L}$. Our objective is to derive closed-form expressions for the first- and second-order directional derivatives of $f$ in the direction of $\boldsymbol{U} \in \mathbb{R}^{K \times L}$, where $\|\boldsymbol{U}\|_F < \infty$, denoted respectively by $D_{\boldsymbol{U}} f(\boldsymbol{X})$ and $D_{\boldsymbol{U}}^2 f(\boldsymbol{X})$. By the limit definition of the derivative, the first derivative of $f(\boldsymbol{X})$ with respect to each entry of $\boldsymbol{X}$ can be expressed as follows:

$$\frac{\partial f(\boldsymbol{X})}{\partial X_{ij}} = \lim_{\Delta t \to 0} \frac{f(\boldsymbol{X} + \Delta t \boldsymbol{e}_i \boldsymbol{e}_j^\top) - f(\boldsymbol{X})}{\Delta t}, \quad \forall (i,j) \in [K] \times [L], \tag{5}$$

where $\boldsymbol{e}_i$ is the $i^{\text{th}}$ standard basis vector of $\mathbb{R}^K$ and $\boldsymbol{e}_j$ is the $j^{\text{th}}$ standard basis vector of $\mathbb{R}^L$. If the limit in (5) exists then by substitution of variables

$$\frac{\partial f(\boldsymbol{X})}{\partial X_{ij}} U_{ij} = \lim_{\Delta t \to 0} \frac{f(\boldsymbol{X} + \Delta t U_{ij} \boldsymbol{e}_i \boldsymbol{e}_j^\top) - f(\boldsymbol{X})}{\Delta t}, \quad \forall (i,j) \in [K] \times [L]. \tag{6}$$

By definition, the total change in $f(\boldsymbol{X})$ in the direction of $\boldsymbol{U}$ is the sum of change due to each entry of $\boldsymbol{X}$. Then

$$D_{\boldsymbol{U}} f(\boldsymbol{X}) = \sum_{i,j \in [K] \times [L]} \frac{\partial f(\boldsymbol{X})}{\partial X_{ij}} U_{ij} \tag{7}$$

$$= \sum_{i,j \in [K] \times [L]} \lim_{\Delta t \to 0} \frac{f(\boldsymbol{X} + \Delta t U_{ij} \boldsymbol{e}_i \boldsymbol{e}_j^\top) - f(\boldsymbol{X})}{\Delta t} \tag{8}$$

$$= \lim_{\Delta t \to 0} \frac{f(\boldsymbol{X} + \Delta t \boldsymbol{U}) - f(\boldsymbol{X})}{\Delta t}. \tag{9}$$

We can rewrite (9) as follows:

$$\lim_{\Delta t \to 0} \frac{f(\boldsymbol{X} + (\Delta t + t)\boldsymbol{U}) - f(\boldsymbol{X} + t\boldsymbol{U})}{\Delta t} \bigg|_{t=0} = \frac{\partial f(\boldsymbol{X} + t\boldsymbol{U})}{\partial t} \bigg|_{t=0}. \tag{10}$$

This is known as the *Gâteaux derivative*, which represents the change in $f(\boldsymbol{X})$ under a perturbation in the direction of $\boldsymbol{U}$. By the same reasoning, we obtain the following result.

**Lemma 2.** *The second directional derivative of $f$ at $\boldsymbol{X}$ in the direction $\boldsymbol{U} \in \mathbb{R}^{K \times L}$ is given by*

$$D_{\boldsymbol{U}}^2 f(\boldsymbol{X}) = \frac{\partial^2}{\partial t^2} f(\boldsymbol{X} + t\boldsymbol{U})\Big|_{t=0}. \tag{11}$$

The proof is deferred to Appendix A.1.

**Directional Second Derivatives and Maximum Eigenvalue.** Consider the following objective function for our real-valued matrix-variable function $f$.

$$f(\boldsymbol{W}_1, \boldsymbol{W}_2, \ldots, \boldsymbol{W}_L) = \|\boldsymbol{M} - \boldsymbol{W}_L \boldsymbol{W}_{L-1} \cdots \boldsymbol{W}_1\|_F^2, \tag{12}$$

where $\boldsymbol{W}_i \in \mathbb{R}^{d_i \times d_{i-1}}$ and $\boldsymbol{M} \in \mathbb{R}^{d_L \times d_0}$ for all $i \in [L]$. In this setting, we can define the largest eigenvalue of the $\nabla^2 f(\boldsymbol{W}_1, \boldsymbol{W}_2, \ldots, \boldsymbol{W}_L)$ at an arbitrary point in the parameter space as follows:

$$\lambda_{\max}(\nabla^2 f(\boldsymbol{W}_1, \boldsymbol{W}_2, \ldots, \boldsymbol{W}_L)) = \max_{\substack{\boldsymbol{U}_1, \boldsymbol{U}_2, \cdots, \boldsymbol{U}_L: \\ \sum_{i=1}^L \|\boldsymbol{U}_i\|_F^2 = 1}} \frac{d^2}{dt^2} f(\boldsymbol{W}_1 + t\boldsymbol{U}_1, \cdots, \boldsymbol{W}_L + t\boldsymbol{U}_L)\Big|_{t=0}. \tag{13}$$

This is the generalization of the Rayleigh quotient to the case where the Hessian is represented as a tensor and its eigenvectors take the form of matrices. This leads to the following lemma.

**Lemma 3.** *For any $[\boldsymbol{W}_1^*, \boldsymbol{W}_2^*, \cdots, \boldsymbol{W}_L^*]$ such that $\boldsymbol{M} = \prod_{j=1}^L \boldsymbol{W}_j^*$, the directional second derivative is given by*

$$\nabla^2 f(\boldsymbol{W}_1^*, \cdots, \boldsymbol{W}_L^*)[\boldsymbol{U}_1, \cdots, \boldsymbol{U}_L] = 2 \left\| \sum_{i=1}^L \left[ \Big( \prod_{j=i+1}^L \boldsymbol{W}_j^* \Big) \boldsymbol{U}_i \Big( \prod_{j=1}^{i-1} \boldsymbol{W}_j^* \Big) \right] \right\|_F^2. \tag{14}$$

The proof is deferred to Appendix A.2.

We study the sharpness of the loss landscape near any global minimum in deep matrix factorization problems. We consider the following optimization problem

$$\min_{\boldsymbol{w} \in \mathbb{R}^N} \mathcal{L}(\boldsymbol{w}) := \|\boldsymbol{M} - \boldsymbol{W}_L \boldsymbol{W}_{L-1} \cdots \boldsymbol{W}_1\|_F^2, \tag{15}$$

where $\boldsymbol{w} = \text{vec}([\boldsymbol{W}_1, \boldsymbol{W}_2, \ldots, \boldsymbol{W}_L])$ denotes the collection of all parameters, and

$$N := \sum_{i=1}^L d_i \times d_{i-1} \tag{16}$$

is the total number of parameters in the model. $\boldsymbol{M} \in \mathbb{R}^{d_L \times d_0}$ denotes the matrix contains the parameters subject to factorization, $L \geq 2$ denotes the depth of factorization and $\boldsymbol{W}_i \in \mathbb{R}^{d_i \times d_{i-1}}$ is the $i^{th}$ factor (layer). This objective is analogous to that of deep linear neural networks. To guarantee the feasibility of factorization at all points in $\mathbb{R}^{d_L \times d_0}$, we require

$$\min_i d_i \geq \min\{d_0, d_L\} \quad \forall i \in [L], \tag{17}$$

which follows directly from the fact that

$$\text{rank}(\boldsymbol{W}_L \boldsymbol{W}_{L-1} \cdots \boldsymbol{W}_1) \leq \min\{\text{rank}(\boldsymbol{W}_1), \text{rank}(\boldsymbol{W}_2), \ldots, \text{rank}(\boldsymbol{W}_L)\}. \tag{18}$$

Define the set of global minima of $\mathcal{L}(\boldsymbol{w})$ as

$$\Omega := \arg\min_{\boldsymbol{w} \in \mathbb{R}^N} \mathcal{L}(\boldsymbol{w}) = \left\{ \boldsymbol{w} \in \mathbb{R}^N : \prod_{i=1}^L \boldsymbol{W}_i = \boldsymbol{M} \right\}. \tag{19}$$

**Local Minima.** Laurent & Brecht (2018) showed that all local minima of deep linear networks with convex and differentiable loss are global if the layers satisfy (17), i.e., hidden layers are at least as wide as input and output layers. In particular, they proved a more general theorem concerning real-valued functions that take as input a product of matrices. Thus, a corollary of Laurent & Brecht (2018, Theorem 1) is that overparameterized deep matrix factorization *does not have spurious minima*, i.e., all local minima are global.

## 3 DEEP OVERPARAMETERIZED SCALAR FACTORIZATION

Before we delve into our general results, we first investigate the deep overparameterized scalar factorization, i.e., a special case of deep matrix factorization in which the first and last layers are vectors. This simplified problem setup reveals the key proof techniques used to prove our general result in Section 4.

**Theorem 4.** *Consider the following objective function*

$$\mathcal{L}(\boldsymbol{w}) := (m - \boldsymbol{w}_L \boldsymbol{W}_{L-1} \cdots \boldsymbol{W}_2 \boldsymbol{w}_1)^2, \tag{20}$$

*where $m \in \mathbb{R}$, $d_0 = d_L = 1$, $\boldsymbol{w}_L \in \mathbb{R}^{1 \times d_{L-1}}$ and $\boldsymbol{w}_1 \in \mathbb{R}^{d_1 \times 1}$. For hidden factors (layers), i.e, for all $i \in \{2, 3, \cdots, L-1\}$, we have $\boldsymbol{W}_i \in \mathbb{R}^{d_i \times d_{i-1}}$. Then, $\forall \boldsymbol{w}^* \in \Omega$,*

$$\lambda_{\max}(\nabla^2 \mathcal{L}(\boldsymbol{w}^*)) = 2 \sum_{i=1}^{L} \sigma_{\max}\Big(\prod_{j=i+1}^{L} \boldsymbol{W}_j^*\Big)^2 \sigma_{\max}\Big(\prod_{j=1}^{i-1} \boldsymbol{W}_j^*\Big)^2. \tag{21}$$

The proof appears in Appendix B.

## 4 OVERPARAMETERIZED DEEP MATRIX FACTORIZATION

We now consider the general deep matrix factorization problem. In this section, we prove our main result, which is closed-form expression for the maximum eigenvalue of the Hessian for any global minimum to the objective (15).

**Theorem 5.** *If $\boldsymbol{w}^* \in \Omega$ then*

$$\lambda_{\max}\big(\nabla^2 \mathcal{L}(\boldsymbol{w}^*)\big) = 2\sigma_{\max}\Big(\sum_{i=1}^{L} \boldsymbol{B}_i^\top \boldsymbol{B}_i \otimes \boldsymbol{A}_i \boldsymbol{A}_i^\top\Big), \tag{22}$$

*where $\boldsymbol{A}_k = \prod_{i=k+1}^{L} \boldsymbol{W}_i^*$ and $\boldsymbol{B}_k = \prod_{i=1}^{k-1} \boldsymbol{W}_i^*$.*

The proof appears in Appendix C.1. Note that the deep overparameterized scalar factorization is a special case of deep matrix factorization where both $\boldsymbol{B}_i^\top \boldsymbol{B}_i$ and $\boldsymbol{A}_i \boldsymbol{A}_i^\top$ reduce to scalars. In that special case, we recover Theorem 4. Another corollary of Theorem 5 is the maximum Hessian eigenvalue for the classical (depth-2) matrix factorization problem. This result may be of independent interest as the expression simplifies considerably.

**Corollary 6.** *Consider the following depth-2 matrix factorization objective*

$$\mathcal{L}(\boldsymbol{L}, \boldsymbol{R}) = \big\| \boldsymbol{M} - \boldsymbol{L}\boldsymbol{R}^\top \big\|_F^2, \tag{23}$$

*where $\boldsymbol{M} \in \mathbb{R}^{d_L \times d_0}$ is the target matrix and $\boldsymbol{L} \in \mathbb{R}^{d_L \times k}$, $\boldsymbol{R} \in \mathbb{R}^{d_0 \times k}$. To ensure the feasibility of the factorization every point in $\mathbb{R}^{d_L \times d_0}$, we choose $k \geq \min\{d_0, d_L\}$. We define the set of minimizers as follows:*

$$\Omega := \arg\min_{\boldsymbol{L}, \boldsymbol{R}} \mathcal{L}(\boldsymbol{L}, \boldsymbol{R}) = \big\{ (\boldsymbol{L}, \boldsymbol{R}) : \boldsymbol{M} = \boldsymbol{L}\boldsymbol{R}^T \big\}. \tag{24}$$

*If $(\boldsymbol{L}, \boldsymbol{R}) \in \Omega$ then*

$$\lambda_{\max}(\nabla^2 \mathcal{L}(\boldsymbol{L}, \boldsymbol{R})) = 2(\sigma_{\max}(\boldsymbol{L})^2 + \sigma_{\max}(\boldsymbol{R})^2). \tag{25}$$

*Proof.* We have from Theorem 5 that

$$\lambda_{\max}(\nabla^2 \mathcal{L}(\boldsymbol{L}, \boldsymbol{R})) = 2\sigma_{\max}(\boldsymbol{I} \otimes \boldsymbol{L}\boldsymbol{L}^\top + \boldsymbol{R}\boldsymbol{R}^\top \otimes \boldsymbol{I}). \tag{26}$$

Using the fact from Horn & Johnson (1994, Theorem 4.4.5), we can write

$$2\sigma_{\max}(\boldsymbol{I} \otimes \boldsymbol{L}\boldsymbol{L}^\top + \boldsymbol{R}\boldsymbol{R}^\top \otimes \boldsymbol{I}) = 2\sigma_{\max}(\boldsymbol{I} \otimes \boldsymbol{L}\boldsymbol{L}^\top) + 2\sigma_{\max}(\boldsymbol{R}\boldsymbol{R}^\top \otimes \boldsymbol{I}). \tag{27}$$

Note that for any matrix $\boldsymbol{A}$ and $\boldsymbol{B}$, $\sigma_{\max}(\boldsymbol{A} \otimes \boldsymbol{B}) = \sigma_{\max}(\boldsymbol{A})\sigma_{\max}(\boldsymbol{B})$, and $\sigma_{\max}(\boldsymbol{A}\boldsymbol{A}^\top) = \sigma_{\max}(\boldsymbol{A}^\top \boldsymbol{A}) = \sigma_{\max}(\boldsymbol{A})^2$. Hence,

$$2(\sigma_{\max}(\boldsymbol{I} \otimes \boldsymbol{L}\boldsymbol{L}^\top) + \sigma_{\max}(\boldsymbol{R}\boldsymbol{R}^\top \otimes \boldsymbol{I})) = 2(\sigma_{\max}(\boldsymbol{L})^2 + \sigma_{\max}(\boldsymbol{R})^2). \tag{28}$$

$\square$

We also provide a self-contained proof of this corollary in Appendix C.2. This result was also recently observed by Josz (2025) independently.

## 5 Flatness

In this section, we reveal remarkable aspects of the loss landscape of general deep matrix factorization problems. First, we show that an optimal solution in deep overparameterized scalar factorization problem is flat if and only if the product of spectral norms of left and right intermediate networks is constant across layers. This is a direct consequence of Theorem 4.

**Corollary 7.** *Consider the following objective function*

$$\mathcal{L}(\boldsymbol{w}) := (m - \boldsymbol{w}_L \boldsymbol{W}_{L-1} \cdots \boldsymbol{W}_2 \boldsymbol{w}_1)^2, \tag{29}$$

*where* $m \in \mathbb{R}$, $d_0 = d_L = 1$, $\boldsymbol{w}_L \in \mathbb{R}^{1 \times d_{L-1}}$ *and* $\boldsymbol{w}_1 \in \mathbb{R}^{d_1 \times 1}$. *For hidden factors (layers), i.e, for all* $i \in \{2, 3, \cdots, L-1\}$, *we have* $\boldsymbol{W}_i \in \mathbb{R}^{d_i \times d_{i-1}}$. *Then,* $\boldsymbol{w}^* \in \Omega_F$ *if and only if*

$$\sigma_{\max}(\boldsymbol{A}_k)\sigma_{\max}(\boldsymbol{B}_k) = |m|^{1-\frac{1}{L}} \quad \forall k \in [L], \tag{30}$$

*where* $\boldsymbol{A}_k = \prod_{i=k+1}^L \boldsymbol{W}_i^*$ *and* $\boldsymbol{B}_k = \prod_{i=1}^{k-1} \boldsymbol{W}_i^*$.

*Proof.* For a minimizer $\boldsymbol{w}^*$, let us assume that $\sigma_{\max}(\boldsymbol{A}_k)\sigma_{\max}(\boldsymbol{B}_k) = |m|^{1-\frac{1}{L}}$ for all $k \in [L]$. Then, by Theorem 4, $\lambda_{\max}(\nabla^2 \mathcal{L}(\boldsymbol{w}^*)) = 2L \times m^{2(1-1/L)}$. This implies that $\boldsymbol{w}^*$ is flat (Mulayoff & Michaeli, 2020, Theorem 1). Now, assume that $\boldsymbol{w}^*$ is flat. (Mulayoff & Michaeli, 2020, Theorem 2) showed that for any flat minimum $\boldsymbol{w}^*$,

$$\sigma_{\max}(\boldsymbol{A}_k) = |m|^{1-\frac{k}{L}} \quad \text{and} \quad \sigma_{\max}(\boldsymbol{B}_k) = |m|^{\frac{k-1}{L}} \quad \forall k \in [L]. \tag{31}$$

Hence, $\sigma_{\max}(\boldsymbol{A}_k)\sigma_{\max}(\boldsymbol{B}_k) = |m|^{1-\frac{1}{L}}$ for all $k \in [L]$. $\qquad \square$

Second, we show that being flat in depth-2 matrix factorization is equivalent to being spectral-norm balanced. This result is a direct consequence of Corollary 6 and AM-GM inequality (Josz, 2025, see also Lemma 5).

**Corollary 8.** *Consider the following depth-2 matrix factorization objective*

$$\mathcal{L}(\boldsymbol{L}, \boldsymbol{R}) = \left\| \boldsymbol{M} - \boldsymbol{L}\boldsymbol{R}^\top \right\|_F^2, \tag{32}$$

*where* $\boldsymbol{M} \in \mathbb{R}^{d_L \times d_0}$ *is the target matrix and* $\boldsymbol{L} \in \mathbb{R}^{d_L \times k}$, $\boldsymbol{R} \in \mathbb{R}^{d_0 \times k}$ *such that* $k \geq \min\{d_0, d_L\}$. *Then,* $(\boldsymbol{L}^*, \boldsymbol{R}^*) \in \Omega_F$ *if and only if*

$$\sigma_{\max}(\boldsymbol{L}^*) = \sigma_{\max}(\boldsymbol{R}^*) = \sqrt{\sigma_{\max}(\boldsymbol{M})}. \tag{33}$$

**Remark 9.** *We discuss further implications of this equivalence in the subsequent section.*

Now, we show that a minimizer of deep matrix factorization loss is flat if the product of spectral norms of left and right intermediate factors is constant across layers.

**Corollary 10.** *Consider the optimization objective in (15). If* $\boldsymbol{w}^* \in \Omega$ *then*

$$\lambda_{\max}(\nabla^2 \mathcal{L}(\boldsymbol{w}^*)) \leq 2 \sum_{k=1}^L \sigma_{\max}(\boldsymbol{A}_k)^2 \sigma_{\max}(\boldsymbol{B}_k)^2, \tag{34}$$

*where* $\boldsymbol{A}_k = \prod_{i=k+1}^L \boldsymbol{W}_i^*$ *and* $\boldsymbol{B}_k = \prod_{i=1}^{k-1} \boldsymbol{W}_i^*$ *(see Appendix B). This implies that if* $\boldsymbol{w}^*$ *satisfies* $\sigma_{\max}(\boldsymbol{A}_k)\sigma_{\max}(\boldsymbol{B}_k) = \sigma_{\max}(\boldsymbol{M})^{1-\frac{1}{L}}$ *for all* $k \in [L]$, *then* $\boldsymbol{w}^*$ *is flat.*

*Proof.* Mulayoff & Michaeli (2020) showed that for any $\boldsymbol{w}^* \in \Omega$,

$$\lambda_{\max}(\nabla^2 \mathcal{L}(\boldsymbol{w}^*)) \geq 2L\sigma_{\max}(\boldsymbol{M})^{2(1-1/L)}. \tag{35}$$

Now, assume that $\sigma_{\max}(\boldsymbol{A}_k)\sigma_{\max}(\boldsymbol{B}_k) = \sigma_{\max}(\boldsymbol{M})^{1-\frac{1}{L}}$ for all $k \in [L]$. Then, by (34),

$$\lambda_{\max}(\nabla^2 \mathcal{L}(\boldsymbol{w}^*)) = 2L\sigma_{\max}(\boldsymbol{M})^{2(1-1/L)}. \tag{36}$$

This implies that $\boldsymbol{w}^*$ is flat (Mulayoff & Michaeli, 2020, Theorem 1). $\qquad \square$

## 6 DISCUSSION

In this section, we thoroughly examine the claims made by Ding et al. (2024) and Ghosh et al. (2025) that flat minima coincide with Frobenius-norm balanced minima in depth-2 matrix factorization and then we discuss how misleading sharpness measures can be for the loss landscape analysis.

**Strict assumptions.** First, we examine the latter work, where Ghosh et al. (2025) examined the learning dynamics of deep linear networks within the deep matrix factorization loss beyond the edge of stability. Their analysis relies on two strict assumptions, which the latter follows the first one. First, they introduce the *singular vector stationary set*, i.e., for any initialization of GD from this set, *GD does not rotate the layers during training*. In Proposition 4, they prove that the balanced initialization they considered in the paper, i.e., $\boldsymbol{W}_L(0) = \boldsymbol{W}_{L-1}(0) = \cdots = \boldsymbol{W}_1(0) = \alpha \boldsymbol{I}$ where $\alpha \in \mathbb{R}$, is a member of the singular vector stationary set. This is leveraged to decouple the dynamics of the singular vectors and singular values. Second, for this specific initialization, singular values remain balanced during training, which is not true for arbitrary initialization. Under these assumptions, their optimization objective becomes a deep scalar factorization problem in which the spectral norm equals the Frobenius norm. Therefore, they claim that *Frobenius-norm balanced minima correspond to flat minima* in deep matrix factorization problem. However, this is only valid from the perspective of GD that initialized at a specific set of points. Globally, we can find minimizers that are **flat but not Frobenius-norm balanced**. To see this, consider a depth-2 matrix factorization problem, i.e.,

$$\min_{\boldsymbol{L},\boldsymbol{R}} \mathcal{L}(\boldsymbol{L}, \boldsymbol{R}) \quad \text{such that} \quad \mathcal{L}(\boldsymbol{L}, \boldsymbol{R}) = \left\| \boldsymbol{M} - \boldsymbol{L}\boldsymbol{R}^\top \right\|_F^2, \tag{37}$$

where $\boldsymbol{M} \in \mathbb{R}^{m \times n}, \boldsymbol{L} \in \mathbb{R}^{m \times k}$ and $\boldsymbol{R} \in \mathbb{R}^{n \times k}$. Suppose $m = n = k = 3$ and $\boldsymbol{M} = \text{diag}(11, 2, 1)$. We know that for any flat minimum $(\boldsymbol{L}', \boldsymbol{R}')$, $\lambda_{\max}(\nabla^2(\mathcal{L}(\boldsymbol{L}', \boldsymbol{R}'))) = 4 \times \sigma_{\max}(\boldsymbol{M})$, which equals 44 (Mulayoff & Michaeli, 2020). By definition, any minimum whose sharpness is equal to 44 is a flat minimum. Let's investigate a specific minimizer $(\boldsymbol{L}^*, \boldsymbol{R}^*)$ such that $\boldsymbol{L}^* = \text{diag}(\sqrt{11}, \frac{2}{3}, 1)$ and $\boldsymbol{R}^* = \text{diag}(\sqrt{11}, 3, 1)$. Using Corollary 6, $\lambda_{\max}(\nabla^2(\mathcal{L}(\boldsymbol{L}^*, \boldsymbol{R}^*))) = 44$, which means that $(\boldsymbol{L}^*, \boldsymbol{R}^*)$ is a flat minimum. On the other hand, by definition of balancedness, i.e., $\boldsymbol{L}\boldsymbol{L}^\top = \boldsymbol{R}\boldsymbol{R}^\top$, $(\boldsymbol{L}^*, \boldsymbol{R}^*)$ is not a balanced minimizer. Therefore, any flat minimum is not necessarily a balanced minimizer.

**Sharpness measures are delusive.** Ding et al. (2024) showed that flat minima recover the groundtruth matrix in low rank matrix recovery. Furthermore, they showed that *norm-minimal*, *balanced*, and *flat* solutions coincide in depth-2 matrix factorization. They measure sharpness of a minimum by using *scaled trace*. Under this measure, flat minima coincide with Frobenius-norm balanced minima; however, as we showed above, this is not necessarily true when the worst-case sharpness, i.e., maximum Hessian eigenvalue, is used as the sharpness measure. In fact, we showed that flat minima are spectral-norm balanced when the maximum Hessian eigenvalue is used as the sharpness measure. This means that a loss landscape analysis in general matrix factorization problems using a specific sharpness measure might not lead to the same inferences as those made by using a different metric.

**The necessity of an absolute sharpness measure.** There is still no consensus in the literature regarding the definition of flatness. For instance, while several works define flat minima as global minimizers that minimize the maximum eigenvalue of the Hessian of the loss (Mulayoff & Michaeli, 2020; Liu et al., 2021; Marion & Chizat, 2024), others define them as global minimizers that minimize the trace of the Hessian (Dinh et al., 2017; Gatmiry et al., 2023). Even a scaled version of the Hessian trace designed to define flat minima in depth-2 matrix factorization problems (Ding et al., 2024), as we mentioned before. As we have shown in Section 5 and discussed above, different sharpness measures can lead to contradictory interpretations of balanced minima. Thus, the necessity of a robust sharpness measure that consolidates the interpretations from existing ones is a matter of urgency. Addressing this concern, Josz (2025) showed how minimizing the maximum Hessian eigenvalue over the solution set can disregard the higher-order variation near minimizers and result in misleading interpretations of flat minima. Therefore, Josz (2025) defined the flat minima of any smooth function $\mathcal{L}$ as the local minima of $\lambda_{\max}(\nabla^2 \mathcal{L}(\boldsymbol{w}))$ under the constraint $\mathcal{L}(\boldsymbol{w}) = \mathcal{L}(\boldsymbol{w}^*)$ and demonstrated that this notion coincides with other notions of flatness in the depth-2 matrix factorization.

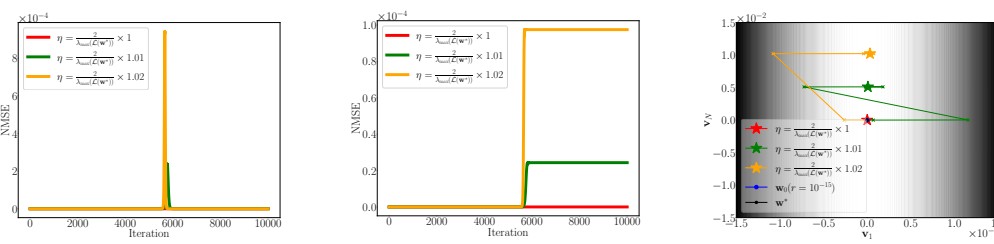

(a) Normalized training error across iterations, i.e., $\mathcal{L}(\boldsymbol{w}_k)/\|\boldsymbol{M}\|_F^2$.

(b) Normalized $\ell^2$ distance of $\boldsymbol{w}_k$ from the minimum $\boldsymbol{w}^*$, i.e, $\|\boldsymbol{w}_k - \boldsymbol{w}^*\|_2^2/\|\boldsymbol{w}^*\|_2^2$.

(c) Trajectories of GD on the contour map of the loss landscape around the minimum.

Figure 1: GD dynamics with different step sizes, $\eta \geq 2/\lambda_{\max}(\nabla^2 \mathcal{L}(\boldsymbol{w}^*))$, indicated by different colors, are initialized within a radius of $10^{-15}$ from the minimum in the direction of the Hessian eigenvector corresponding to the largest eigenvalue, for depth-2 matrix factorization, $\boldsymbol{M} = \boldsymbol{L}\boldsymbol{R}^\top$, of a random Gaussian matrix, where $\boldsymbol{L} \in \mathbb{R}^{10 \times 20}$ and $\boldsymbol{R} \in \mathbb{R}^{20 \times 20}$. The vector $\boldsymbol{v}_1$ denotes the eigenvector of the Hessian corresponding to the largest eigenvalue, while $\boldsymbol{v}_N$ denotes the eigenvector corresponding to the smallest eigenvalue. The value of $\lambda_{\max}(\nabla^2 \mathcal{L}(\boldsymbol{w}^*))$ is computed using the closed-form expression derived in Corollary 6.

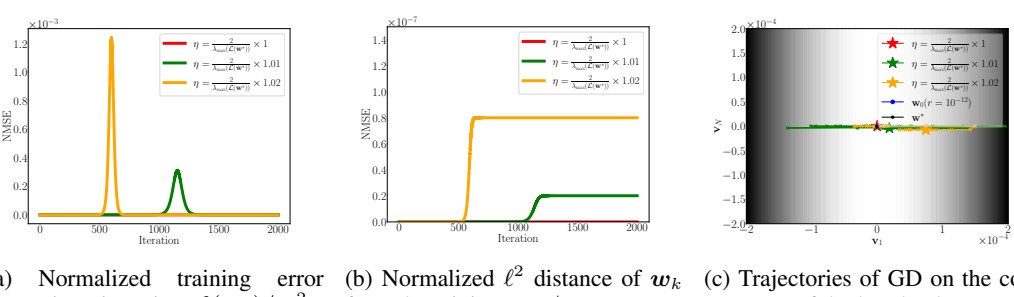

(a) Normalized training error across iterations, i.e., $\mathcal{L}(\boldsymbol{w}_k)/m^2$.

(b) Normalized $\ell^2$ distance of $\boldsymbol{w}_k$ from the minimum $\boldsymbol{w}^*$.

(c) Trajectories of GD on the contour map of the loss landscape.

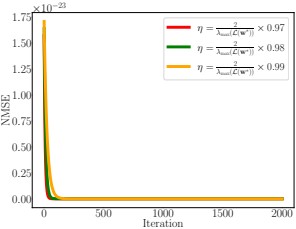 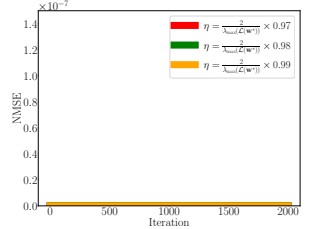 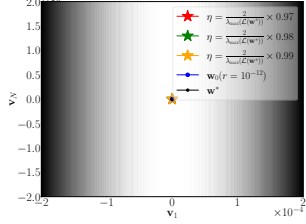

(d) Normalized training error across iterations, i.e., $\mathcal{L}(\boldsymbol{w}_k)/m^2$.

(e) Normalized $\ell^2$ distance of $\boldsymbol{w}_k$ from the minimum $\boldsymbol{w}^*$.

(f) Trajectories of GD on the contour map of the loss landscape.

Figure 2: GD dynamics with different step sizes indicated by different colors are initialized within a radius of $10^{-12}$ from the minimum in the direction of the Hessian eigenvector corresponding to the largest eigenvalue, for a 15-layer overparameterized scalar factorization of a random scalar. The value of $\lambda_{\max}(\nabla^2 \mathcal{L}(\boldsymbol{w}^*))$ is computed using the closed-form expression derived in Theorem 4.

## 7 EXPERIMENTS

The dynamical stability analysis relies on how accurate the quadratic approximation of the loss function is in the $\delta$-neighborhood of $\boldsymbol{x}^*$ (Wu et al., 2018) (also see Definition 1). Therefore, we initialize GD in a $\delta$-neighborhood with $\delta$ on the order of $10^{-15}$ to $10^{-9}$ to ensure validity of the quadratic approximation. To observe escape phenomenon clearly, we choose the perturbation direction for the initial point to be the eigenvector of $\nabla^2 \mathcal{L}(\boldsymbol{w}^*)$ corresponding to the largest eigenvalue, so as to avoid choosing a direction that is orthogonal to the eigenspace corresponding to the eigenvalues larger than $2/\eta$.

Note that, we could have also considered a random perturbation direction uniformly from $\mathbb{S}^{N-1}$. Mulayoff & Michaeli (2020) showed that for the deep matrix factorization problem (which has the same Hessian structure with deep linear networks), the Hessian is rank-deficient at all minima by at least the order of $1 - (1/L)$. This means that at least $1 - (1/L)$ of the eigenvalues are zero at a minimum. Thus, the probability of choosing a direction that is orthogonal to the eigenspace corresponding to eigenvalues larger than $2/\eta$ is 0. Therefore, random perturbations would lead to the escape phenomenon with probability 1.

On the other hand, for stable minima, it is important to choose the direction as the eigenvector of $\nabla^2 \mathcal{L}(\boldsymbol{w}^*)$ corresponding to the largest eigenvalue. The reasoning is the same. If you choose a direction that is not orthogonal to the eigenspace corresponding to the eigenvalues that are zero, then GD never converges to $\boldsymbol{w}^*$. This means that if we choose the perturbation direction randomly, then with probability 1, we choose a direction that is not orthogonal to the eigenspace corresponding to the eigenvalues that are zero. Therefore, for the experiment, it is convenient to choose perturbation direction as the eigenvector of $\nabla^2 \mathcal{L}(\boldsymbol{w}^*)$ corresponding to the largest eigenvalue.

To measure the distance between the convergence point and the minimizer, we plot the normalized $\ell^2$-norm of $\boldsymbol{w}_k - \boldsymbol{w}^*$ at each iteration. Furthermore, as shown in Fig. 1 and Figs. 2a-2c if $\eta > 2/\lambda_{\max}$, where $\lambda_{\max} := \lambda_{\max}(\nabla^2 \mathcal{L}(\boldsymbol{w}^*))$, GD always escapes from the minimum. On the other hand, if $\eta = 2/\lambda_{\max}$ then GD converges as shown in Figs. 2d-2f. A catapult in the training error indicates GD's escape from the basin of a minimum, after which it eventually converges to another minimum as observed by Wu et al. (2018). For the methodology used to generate contour maps of the loss landscape near a minimum, see Appendix D.1, and for additional experiments, see Appendix D.2.

## 8 CONCLUSION

In this paper, we derived an exact expression for the maximum eigenvalue of the Hessian of the squared-error loss for overparameterized deep matrix factorization problems at any minimizer. We also showed that this expression simplifies considerably for depth-2 matrix factorization and deep, overparameterized scalar factorization. To complement our theory, we conducted GD experiments that crucially rely on our exact expression of the sharpness to observe the escape phenomenon during training. Then, we showed how flat minima are spectral-norm balanced in depth-2 matrix factorization, and we showed that a minimizer of deep overparameterized scalar factorization loss is flat if and only if the product of spectral norms of left and right intermediate factors is constant across layers. Furthermore, we showed that a minimizer of deep matrix factorization loss is flat if the product of spectral norms of left and right intermediate factors is constant across layers. These results led to the following remarkable observation: flat minima are not necessarily Frobenius-norm balanced, which is contradictory to the claim made in (Ghosh et al., 2025). Finally, we discussed how delusive the sharpness measures can be for the loss landscape analysis. Therefore, there is an urgent need for future research to explore a robust sharpness measure that consolidates the interpretations from existing ones.

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

# A    PROOFS FROM SECTION 2

## A.1    PROOF OF LEMMA 2

*Proof.* We can express the derivative of $\frac{\partial f(\boldsymbol{X})}{\partial X_{ij}}$ with respect to each entry of $\boldsymbol{X}$, using the limit definition of the derivative, as follows:

$$\frac{\partial^2 f(\boldsymbol{X})}{\partial X_{kl}\partial X_{ij}} = \frac{\partial}{\partial X_{kl}}\left(\frac{\partial f(\boldsymbol{X})}{\partial X_{ij}}\right) = \lim_{\Delta t \to 0}\frac{\partial f(\boldsymbol{X} + \Delta t \boldsymbol{e}_i \boldsymbol{e}_j^\top) - \partial f(\boldsymbol{X})}{\partial X_{kl}\Delta t}, \quad \forall (k,l) \in [K] \times [L] \tag{38}$$

which is equal to

$$\lim_{\Delta h, \Delta t \to 0}\frac{f(\boldsymbol{X} + \Delta t \boldsymbol{e}_i \boldsymbol{e}_j^\top + \Delta h \boldsymbol{e}_k \boldsymbol{e}_l^\top) - f(\boldsymbol{X} + \Delta t \boldsymbol{e}_i \boldsymbol{e}_j^\top) - f(\boldsymbol{X} + \Delta h \boldsymbol{e}_k \boldsymbol{e}_l^\top) + f(\boldsymbol{X})}{\Delta h \Delta t}. \tag{39}$$

By using the substitution of variables as in (6),

$$\frac{\partial^2 f(\boldsymbol{X})}{\partial X_{kl}\partial X_{ij}}U_{ij}U_{kl} = \frac{\partial}{\partial X_{kl}}\left(\frac{\partial f(\boldsymbol{X})}{\partial X_{ij}}U_{ij}\right)U_{kl} = \lim_{\Delta t \to 0}\frac{\partial f(\boldsymbol{X} + \Delta t U_{ij}\boldsymbol{e}_i \boldsymbol{e}_j^\top) - \partial f(\boldsymbol{X})}{\partial X_{kl}\Delta t}U_{kl}. \tag{40}$$

Equivalently,

$$\lim_{\Delta h, \Delta t \to 0}\frac{f(\boldsymbol{X} + \Delta t U_{ij}\boldsymbol{e}_i \boldsymbol{e}_j^\top + \Delta h U_{kl}\boldsymbol{e}_k \boldsymbol{e}_l^\top) - f(\boldsymbol{X} + \Delta t U_{ij}\boldsymbol{e}_i \boldsymbol{e}_j^\top) - f(\boldsymbol{X} + \Delta h U_{kl}\boldsymbol{e}_k \boldsymbol{e}_l^\top) + f(\boldsymbol{X})}{\Delta h \Delta t}. \tag{41}$$

which can be proved by substitution of variables in (40). In turn, second order differential due to any $\boldsymbol{U} \in \mathbb{R}^{K \times L}$ is

$$D_{\boldsymbol{U}}^2 f(\boldsymbol{X}) = \sum_{i,j}\sum_{k,l}\frac{\partial^2 f(\boldsymbol{X})}{\partial X_{kl}\partial X_{ij}}U_{ij}U_{kl} = \left\langle \nabla \langle \nabla f(\boldsymbol{X}), \boldsymbol{U}\rangle, \boldsymbol{U}\right\rangle, \tag{42}$$

where

$$\nabla f(\boldsymbol{X}) = \begin{bmatrix} \frac{\partial f(\boldsymbol{X})}{\partial X_{11}} & \frac{\partial f(\boldsymbol{X})}{\partial X_{12}} & \cdots & \frac{\partial f(\boldsymbol{X})}{\partial X_{1L}} \\ \frac{\partial f(\boldsymbol{X})}{\partial X_{21}} & \frac{\partial f(\boldsymbol{X})}{\partial X_{22}} & \cdots & \frac{\partial f(\boldsymbol{X})}{\partial X_{2L}} \\ \vdots & \ddots & \cdots & \vdots \\ \frac{\partial f(\boldsymbol{X})}{\partial X_{K1}} & \frac{\partial f(\boldsymbol{X})}{\partial X_{K2}} & \cdots & \frac{\partial f(\boldsymbol{X})}{\partial X_{KL}} \end{bmatrix} \in \mathbb{R}^{K \times L}. \tag{43}$$

Equivalently,

$$D_{\boldsymbol{U}}^2 f(\boldsymbol{X}) = \sum_{k,l}\lim_{\Delta t \to 0}\frac{\partial f(\boldsymbol{X} + \Delta t \boldsymbol{U}\boldsymbol{e}_i \boldsymbol{e}_j^\top) - \partial f(\boldsymbol{X})}{\partial X_{kl}\Delta t}U_{kl} \tag{44}$$

$$= \lim_{\Delta t \to 0}\frac{f(\boldsymbol{X} + 2\Delta t \boldsymbol{U}) - 2f(\boldsymbol{X} + \Delta t \boldsymbol{U}) + f(\boldsymbol{X})}{\Delta t^2} \tag{45}$$

$$= \frac{\partial^2}{\partial t^2}f(\boldsymbol{X} + t\boldsymbol{U})\bigg|_{t=0}. \tag{46}$$

$\square$

## A.2    PROOF OF LEMMA 3

*Proof.* We can rewrite (12) through using the definition of *Frobenius inner product*

$$f(\boldsymbol{W}_1, \cdots, \boldsymbol{W}_L) = \left\langle \boldsymbol{M} - \prod_{i=1}^L \boldsymbol{W}_i, \boldsymbol{M} - \prod_{i=1}^L \boldsymbol{W}_i\right\rangle. \tag{47}$$

Then

$$f(\boldsymbol{W}_1 + t\boldsymbol{U}_1, \cdots, \boldsymbol{W}_L + t\boldsymbol{U}_L) = \left\langle \boldsymbol{M} - \prod_{i=1}^{L}(\boldsymbol{W}_i + t\boldsymbol{U}_i), \boldsymbol{M} - \prod_{i=1}^{L}(\boldsymbol{W}_i + t\boldsymbol{U}_i) \right\rangle. \tag{48}$$

Let's define $g : \mathbb{R} \longrightarrow \mathbb{R}^{d_L \times d_0}$ such that

$$g(t) = \boldsymbol{M} - \prod_{i=1}^{L}(\boldsymbol{W}_i + t\boldsymbol{U}_i), \quad f(\boldsymbol{W}_1 + t\boldsymbol{U}_1, \cdots, \boldsymbol{W}_L + t\boldsymbol{U}_L) = \left\langle g(t), g(t) \right\rangle. \tag{49}$$

First, we need to differentiate $f(\boldsymbol{W}_1 + t\boldsymbol{U}_1, \cdots, \boldsymbol{W}_L + t\boldsymbol{U}_L)$ w.r.t $t$. Using the fact that $\left\langle \boldsymbol{A}, \boldsymbol{B} \right\rangle = \operatorname{tr}(\boldsymbol{A}^\top \boldsymbol{B})$, which simplifies the differentiation,

$$\frac{d}{dt} f(\boldsymbol{W}_1 + t\boldsymbol{U}_1, \cdots, \boldsymbol{W}_L + t\boldsymbol{U}_L) = 2\left\langle g(t), g'(t) \right\rangle. \tag{50}$$

$$\frac{d^2}{dt^2} f(\boldsymbol{W}_1 + t\boldsymbol{U}_1, \cdots, \boldsymbol{W}_L + t\boldsymbol{U}_L) = 2\left\langle g'(t), g'(t) \right\rangle + 2\left\langle g(t), g''(t) \right\rangle. \tag{51}$$

Then, the directional second derivative $\nabla^2 f(\boldsymbol{W}_1, \cdots, \boldsymbol{W}_L)[\boldsymbol{U}_1, \cdots, \boldsymbol{U}_L]$ equals

$$\frac{d^2}{dt^2} f(\boldsymbol{W}_1 + t\boldsymbol{U}_1, \cdots, \boldsymbol{W}_L + t\boldsymbol{U}_L)\Big|_{t=0} = 2\left\langle g'(0), g'(0) \right\rangle + 2\left\langle g(0), g''(0) \right\rangle. \tag{52}$$

It is straightforward to differentiate $g(t)$ such that

$$g(0) = \boldsymbol{M} - \prod_{i=1}^{L} \boldsymbol{W}_i, \tag{53}$$

$$g'(0) = -\sum_{i=1}^{L}\left[\left(\prod_{j=i+1}^{L} \boldsymbol{W}_j\right)\boldsymbol{U}_i\left(\prod_{j=1}^{i-1} \boldsymbol{W}_j\right)\right], \tag{54}$$

$$g''(0) = -2\sum_{1 \le k < i \le L}\left[\left(\prod_{j=i+1}^{L} \boldsymbol{W}_j\right)\boldsymbol{U}_i\left(\prod_{j=k+1}^{i-1} \boldsymbol{W}_j\right)\boldsymbol{U}_k\left(\prod_{j=1}^{k-1} \boldsymbol{W}_j\right)\right]. \tag{55}$$

Therefore, for any $[\boldsymbol{W}_1, \boldsymbol{W}_2, \cdots, \boldsymbol{W}_L]$ in parameter space

$$\nabla^2 f(\boldsymbol{W}_1, \cdots, \boldsymbol{W}_L)[\boldsymbol{U}_1, \cdots, \boldsymbol{U}_L] = \tag{56}$$

$$2\left\langle \sum_{i=1}^{L}\left[\left(\prod_{j=i+1}^{L} \boldsymbol{W}_j\right)\boldsymbol{U}_i\left(\prod_{j=1}^{i-1} \boldsymbol{W}_j\right)\right], \sum_{i=1}^{L}\left[\left(\prod_{j=i+1}^{L} \boldsymbol{W}_j\right)\boldsymbol{U}_i\left(\prod_{j=1}^{i-1} \boldsymbol{W}_j\right)\right] \right\rangle \tag{57}$$

$$-4\left\langle \boldsymbol{M} - \prod_{i=1}^{L} \boldsymbol{W}_i, \sum_{1 \le k < i \le L}\left[\left(\prod_{j=i+1}^{L} \boldsymbol{W}_j\right)\boldsymbol{U}_i\left(\prod_{j=k+1}^{i-1} \boldsymbol{W}_j\right)\boldsymbol{U}_k\left(\prod_{j=1}^{k-1} \boldsymbol{W}_j\right)\right] \right\rangle. \tag{58}$$

Note that for any minimizer $[\boldsymbol{W}_1^*, \boldsymbol{W}_2^*, \cdots, \boldsymbol{W}_L^*]$, $\boldsymbol{M} - \prod_{j=1}^{L} \boldsymbol{W}_i^* = 0$. Hence, for any global minimum

$$\nabla^2 f(\boldsymbol{W}_1^*, \cdots, \boldsymbol{W}_L^*)[\boldsymbol{U}_1, \cdots, \boldsymbol{U}_L] = \tag{59}$$

$$2\left\langle \sum_{i=1}^{L}\left[\left(\prod_{j=i+1}^{L} \boldsymbol{W}_j^*\right)\boldsymbol{U}_i\left(\prod_{j=1}^{i-1} \boldsymbol{W}_j^*\right)\right], \sum_{i=1}^{L}\left[\left(\prod_{j=i+1}^{L} \boldsymbol{W}_j^*\right)\boldsymbol{U}_i\left(\prod_{j=1}^{i-1} \boldsymbol{W}_j^*\right)\right] \right\rangle. \tag{60}$$

$\square$

# B  WARM-UP: DEEP OVERPARAMETERIZED SCALAR FACTORIZATION

*Proof.* According to (13) and (14),

$$\lambda_{\max}\big(\nabla^2\mathcal{L}(\boldsymbol{w}^*)\big) = \max_{\substack{\boldsymbol{U}_1,\boldsymbol{U}_2,\ldots,\boldsymbol{U}_L: \\ \sum_{i=1}^{L}\|\boldsymbol{U}_i\|_F^2=1}} 2\left\|\sum_{i=1}^{L}\left[\Big(\prod_{j=i+1}^{L}\boldsymbol{W}_j^*\Big)\boldsymbol{U}_i\Big(\prod_{j=1}^{i-1}\boldsymbol{W}_j^*\Big)\right]\right\|_F^2 \tag{61}$$

$$\leq \max_{\substack{\boldsymbol{U}_1,\boldsymbol{U}_2,\ldots,\boldsymbol{U}_L: \\ \sum_{i=1}^{L}\|\boldsymbol{U}_i\|_F^2=1}} 2\left(\sum_{i=1}^{L}\left\|\Big(\prod_{j=i+1}^{L}\boldsymbol{W}_j^*\Big)\boldsymbol{U}_i\Big(\prod_{j=1}^{i-1}\boldsymbol{W}_j^*\Big)\right\|_F\right)^2 \tag{62}$$

$$= \max_{\substack{\boldsymbol{u}_1,\boldsymbol{u}_2,\ldots,\boldsymbol{u}_L: \\ \sum_{i=1}^{L}\|\boldsymbol{u}_i\|_2^2=1}} 2\left(\sum_{i=1}^{L}\left\|\left[\Big(\prod_{j=1}^{i-1}\boldsymbol{W}_j^*\Big)^{\top}\otimes\Big(\prod_{j=i+1}^{L}\boldsymbol{W}_j^*\Big)\right]\boldsymbol{u}_i\right\|_2\right)^2 \tag{63}$$

$$\leq \max_{\substack{\boldsymbol{u}_1,\boldsymbol{u}_2,\ldots,\boldsymbol{u}_L: \\ \sum_{i=1}^{L}\|\boldsymbol{u}_i\|_2^2=1}} 2\left(\sum_{i=1}^{L}\sigma_{\max}\left(\Big(\prod_{j=1}^{i-1}\boldsymbol{W}_j^*\Big)^{\top}\otimes\Big(\prod_{j=i+1}^{L}\boldsymbol{W}_j^*\Big)\right)\|\boldsymbol{u}_i\|_2\right)^2 \tag{64}$$

$$= \max_{\substack{\boldsymbol{u}_1,\boldsymbol{u}_2,\ldots,\boldsymbol{u}_L: \\ \sum_{i=1}^{L}\|\boldsymbol{u}_i\|_2^2=1}} 2\left(\sum_{i=1}^{L}\sigma_{\max}\Big(\prod_{j=i+1}^{L}\boldsymbol{W}_j^*\Big)\sigma_{\max}\Big(\prod_{j=1}^{i-1}\boldsymbol{W}_j^*\Big)\|\boldsymbol{u}_i\|_2\right)^2, \tag{65}$$

where $\boldsymbol{U}_i \in \mathbb{R}^{d_i\times d_{i-1}}$ and $\boldsymbol{u}_i \in \mathbb{R}^{d_i d_{i-1}}$. We can upper bound the right-hand side of (61) by using the *triangle inequality*. By applying the *vectorization trick* of the Kronecker product, we can rewrite (62). Then, noting the fact that for any matrix $\boldsymbol{A} \in \mathbb{R}^{m\times n}$ and vector $\boldsymbol{x} \in \mathbb{R}^{n}$, $\|\boldsymbol{A}\boldsymbol{x}\|_2 \leq \sigma_{\max}(\boldsymbol{A})\|\boldsymbol{x}\|_2$, we can upper bound the right-hand side of (63). Note that for any matrix $\boldsymbol{A}$ and $\boldsymbol{B}$, $\sigma_{\max}(\boldsymbol{A}\otimes\boldsymbol{B}) = \sigma_{\max}(\boldsymbol{A})\sigma_{\max}(\boldsymbol{B})$. Hence, we can rewrite (64). Then, by using the Cauchy–Schwarz inequality,

$$\lambda_{\max}\big(\nabla^2\mathcal{L}(\boldsymbol{w}^*)\big) \leq 2\sum_{i=1}^{L}\sigma_{\max}\Big(\prod_{j=i+1}^{L}\boldsymbol{W}_j^*\Big)^2\sigma_{\max}\Big(\prod_{j=1}^{i-1}\boldsymbol{W}_j^*\Big)^2. \tag{66}$$

Then, it suffices to show that there exists a direction $[\boldsymbol{U}_1^*,\boldsymbol{U}_2^*,\ldots,\boldsymbol{U}_L^*]$ along which the bound in (66) is achieved.

Consider decomposition of $\prod_{j=i+1}^{L}\boldsymbol{W}_j^*$ by SVD, and denote by $\boldsymbol{u}_{L_i}$ and $\boldsymbol{v}_{L_i}$ the left and right singular vectors of $\prod_{j=i+1}^{L}\boldsymbol{W}_j^*$ corresponding to the largest singular value, respectively. Note that since $\boldsymbol{W}_L$ is a vector, we have $\boldsymbol{u}_{L_i} = 1$ for all $i \in [L]$. Moreover, decompose $\prod_{j=1}^{i-1}\boldsymbol{W}_j^*$ by SVD, and denote by $\boldsymbol{u}_{R_i}$ and $\boldsymbol{v}_{R_i}$ the left and right singular vectors of $\prod_{j=1}^{i-1}\boldsymbol{W}_j^*$ corresponding to the largest singular value, respectively. Note that since $\boldsymbol{W}_1$ is a vector, we have $\boldsymbol{v}_{R_i} = 1$ for all $i \in [L]$. Now, we determine a particular direction $[\boldsymbol{U}_1^*,\boldsymbol{U}_2^*,\ldots,\boldsymbol{U}_L^*]$ such that they achieve the upper bound while satisfying the constraint $\sum_{i=1}^{L}\|\boldsymbol{U}_i^*\|_F^2 = 1$. Choose

$$\boldsymbol{U}_i^* = \frac{\sigma_{\max}\Big(\prod_{j=i+1}^{L}\boldsymbol{W}_j^*\Big)\sigma_{\max}\Big(\prod_{j=1}^{i-1}\boldsymbol{W}_j^*\Big)}{\sqrt{\sum_{i=1}^{L}\sigma_{\max}\Big(\prod_{j=i+1}^{L}\boldsymbol{W}_j^*\Big)^2\sigma_{\max}\Big(\prod_{j=1}^{i-1}\boldsymbol{W}_j^*\Big)^2}}\boldsymbol{v}_{L_i}\boldsymbol{u}_{R_i}^{\top}. \tag{67}$$

Then,

$$2\left\|\sum_{i=1}^{L}\left[\Big(\prod_{j=i+1}^{L}\boldsymbol{W}_j^*\Big)\boldsymbol{U}_i^*\Big(\prod_{j=1}^{i-1}\boldsymbol{W}_j^*\Big)\right]\right\|_F^2 = 2\sum_{i=1}^{L}\sigma_{\max}\Big(\prod_{j=i+1}^{L}\boldsymbol{W}_j^*\Big)^2\sigma_{\max}\Big(\prod_{j=1}^{i-1}\boldsymbol{W}_j^*\Big)^2. \tag{68}$$

Since the upper bound is achieved, it implies

$$\lambda_{\max}(\nabla^2\mathcal{L}(\boldsymbol{w}^*)) = 2\sum_{i=1}^{L}\sigma_{\max}\Big(\prod_{j=i+1}^{L}\boldsymbol{W}_j^*\Big)^2\sigma_{\max}\Big(\prod_{j=1}^{i-1}\boldsymbol{W}_j^*\Big)^2. \tag{69}$$

$\square$

## C  PROOFS FROM SECTION 4

### C.1  PROOF OF THEOREM 5

*Proof.* By definition,

$$\lambda_{\max}(\nabla^2 \mathcal{L}(\boldsymbol{w}^*)) = \max_{\substack{\boldsymbol{U}_1, \boldsymbol{U}_2, \ldots, \boldsymbol{U}_L: \\ \sum_{i=1}^{L} \|\boldsymbol{U}_i\|_F^2 = 1}} 2 \left\| \sum_{i=1}^{L} \left[ \left( \prod_{j=i+1}^{L} \boldsymbol{W}_j^* \right) \boldsymbol{U}_i \left( \prod_{j=1}^{i-1} \boldsymbol{W}_j^* \right) \right] \right\|_F^2 \tag{70}$$

$$= \max_{\substack{\boldsymbol{U}_1, \boldsymbol{U}_2, \ldots, \boldsymbol{U}_L: \\ \sum_{i=1}^{L} \|\boldsymbol{U}_i\|_F^2 = 1}} 2 \left\| \mathrm{vec} \left( \sum_{i=1}^{L} \left[ \left( \prod_{j=i+1}^{L} \boldsymbol{W}_j^* \right) \boldsymbol{U}_i \left( \prod_{j=1}^{i-1} \boldsymbol{W}_j^* \right) \right] \right) \right\|_2^2 \tag{71}$$

$$= \max_{\substack{\boldsymbol{U}_1, \boldsymbol{U}_2, \ldots, \boldsymbol{U}_L: \\ \sum_{i=1}^{L} \|\boldsymbol{U}_i\|_F^2 = 1}} 2 \left\| \sum_{i=1}^{L} \mathrm{vec} \left( \left( \prod_{j=i+1}^{L} \boldsymbol{W}_j^* \right) \boldsymbol{U}_i \left( \prod_{j=1}^{i-1} \boldsymbol{W}_j^* \right) \right) \right\|_2^2 \tag{72}$$

$$= \max_{\substack{\boldsymbol{u}_1, \boldsymbol{u}_2, \ldots, \boldsymbol{u}_L: \\ \sum_{i=1}^{L} \|\boldsymbol{u}_i\|_2^2 = 1}} 2 \left\| \sum_{i=1}^{L} \left[ \left( \prod_{j=1}^{i-1} \boldsymbol{W}_j^* \right)^\top \otimes \left( \prod_{j=i+1}^{L} \boldsymbol{W}_j^* \right) \right] \boldsymbol{u}_i \right\|_2^2. \tag{73}$$

Note that vec is a linear operator. Therefore, (71) can be rewritten as (72). Then, by using the vectorization trick of the Kronecker product, we can obtain (73). Let's define a block matrix and a vector such that

$$\boldsymbol{K} = \left[ \boldsymbol{I} \otimes \prod_{j=2}^{L} \boldsymbol{W}_j^* | \boldsymbol{W}_1^{*\top} \otimes \left( \prod_{j=3}^{L} \boldsymbol{W}_j^* \right) | \quad \cdots \quad | \left( \prod_{j=1}^{L-1} \boldsymbol{W}_j^* \right)^\top \otimes \boldsymbol{I} \right], \tag{74}$$

$$\boldsymbol{u} = [\boldsymbol{u}_1^\top \quad \boldsymbol{u}_2^\top \quad \cdots \quad \boldsymbol{u}_L^\top]^\top. \tag{75}$$

Then,

$$\lambda_{\max}(\nabla^2 \mathcal{L}(\boldsymbol{w}^*)) = \max_{\boldsymbol{u}: \|\boldsymbol{u}\|_2 = 1} 2 \|\boldsymbol{K} \boldsymbol{u}\|_2^2 \tag{76}$$

$$= \sigma_{\max}(\boldsymbol{K}^\top \boldsymbol{K}). \tag{77}$$

Note that $\sigma_{\max}(\boldsymbol{K}^\top \boldsymbol{K}) = \sigma_{\max}(\boldsymbol{K}\boldsymbol{K}^\top)$, and for any two block matrices $\boldsymbol{A}$ and $\boldsymbol{B}$ such that

$$\boldsymbol{A} = [\boldsymbol{A}_1 \quad \boldsymbol{A}_2 \quad \ldots \quad \boldsymbol{A}_L] \in \mathbb{R}^{M_1 \times d}, \quad \boldsymbol{B} = \begin{bmatrix} \boldsymbol{B}_1 \\ \vdots \\ \boldsymbol{B}_L \end{bmatrix} \in \mathbb{R}^{d \times M_2} \tag{78}$$

$$\boldsymbol{A}\boldsymbol{B} = \sum_{i=1}^{L} \boldsymbol{A}_i \boldsymbol{B}_i, \quad \boldsymbol{A}\boldsymbol{B} \in \mathbb{R}^{M_1 \times M_2}. \tag{79}$$

Furthermore, for any matrices $\boldsymbol{A}, \boldsymbol{B}, \boldsymbol{C}, \boldsymbol{D}$ such that the matrix products $\boldsymbol{A}\boldsymbol{B}$ and $\boldsymbol{C}\boldsymbol{D}$ are well defined, we have

$$(\boldsymbol{A} \otimes \boldsymbol{C})(\boldsymbol{B} \otimes \boldsymbol{D}) = \boldsymbol{A}\boldsymbol{B} \otimes \boldsymbol{C}\boldsymbol{D}. \tag{80}$$

Using the fact that $(\boldsymbol{A} \otimes \boldsymbol{C})^\top = \boldsymbol{A}^\top \otimes \boldsymbol{C}^\top$ together with the previous property, it follows that

$$\lambda_{\max}(\nabla^2 \mathcal{L}(\boldsymbol{w}^*)) = 2\sigma_{\max} \left( \sum_{i=1}^{L} \boldsymbol{B}_i^\top \boldsymbol{B}_i \otimes \boldsymbol{A}_i \boldsymbol{A}_i^\top \right), \tag{81}$$

where $\boldsymbol{A}_k = \prod_{i=k+1}^{L} \boldsymbol{W}_i^*$ and $\boldsymbol{B}_k = \prod_{i=1}^{k-1} \boldsymbol{W}_i^*$.

$\square$

## C.2 Proof of Corollary 6

*Proof.* According to (14), for any $(\boldsymbol{L}^*, \boldsymbol{R}^*) \in \Omega$,

$$\nabla^2 \mathcal{L}(\boldsymbol{L}^*, \boldsymbol{R}^*)[\boldsymbol{U}, \boldsymbol{V}] = 2\big\|\boldsymbol{L}^*\boldsymbol{U}^\top + \boldsymbol{V}\boldsymbol{R}^{*\top}\big\|_F^2. \tag{82}$$

Then,

$$\lambda_{\max}(\nabla^2 \mathcal{L}(\boldsymbol{L}^*, \boldsymbol{R}^*)) = \max_{\substack{\boldsymbol{U}, \boldsymbol{V} \\ \|\boldsymbol{U}\|_F^2 + \|\boldsymbol{V}\|_F^2 = 1}} 2\big\|\boldsymbol{L}^*\boldsymbol{U}^\top + \boldsymbol{V}\boldsymbol{R}^{*\top}\big\|_F^2 \tag{83}$$

$$\leq \max_{\substack{\boldsymbol{U}, \boldsymbol{V} \\ \|\boldsymbol{U}\|_F^2 + \|\boldsymbol{V}\|_F^2 = 1}} 2\big(\big\|\boldsymbol{L}^*\boldsymbol{U}^\top\big\|_F + \big\|\boldsymbol{V}\boldsymbol{R}^{*\top}\big\|_F\big)^2 \tag{84}$$

$$= \max_{\substack{\boldsymbol{u}, \boldsymbol{v} \\ \|\boldsymbol{u}\|_2^2 + \|\boldsymbol{v}\|_2^2 = 1}} 2\Big(\|(\boldsymbol{I} \otimes \boldsymbol{L}^*)\boldsymbol{u}\|_2 + \|(\boldsymbol{R}^* \otimes \boldsymbol{I})\boldsymbol{v}\|_2\Big)^2 \tag{85}$$

$$\leq \max_{\substack{\boldsymbol{u}, \boldsymbol{v} \\ \|\boldsymbol{u}\|_2^2 + \|\boldsymbol{v}\|_2^2 = 1}} 2\Big(\sigma_{\max}(\boldsymbol{I} \otimes \boldsymbol{L}^*)\|\boldsymbol{u}\|_2 + \sigma_{\max}(\boldsymbol{R}^* \otimes \boldsymbol{I})\|\boldsymbol{v}\|_2\Big)^2. \tag{86}$$

We can upper bound the right-hand side of (83) using the *triangle inequality*. By applying the *vectorization trick* of the Kronecker product again, we can rewrite (84). Then, noting that for any matrix $\boldsymbol{A} \in \mathbb{R}^{m \times n}$ and vector $\boldsymbol{x} \in \mathbb{R}^n$, $\|\boldsymbol{A}\boldsymbol{x}\|_2 \leq \sigma_{\max}(\boldsymbol{A})\|\boldsymbol{x}\|_2$, we can upper bound the right-hand side of (85). Note that for any matrix $\boldsymbol{A}$ and $\boldsymbol{B}$, $\sigma_{\max}(\boldsymbol{A} \otimes \boldsymbol{B}) = \sigma_{\max}(\boldsymbol{A})\sigma_{\max}(\boldsymbol{B})$. Hence,

$$\lambda_{\max}(\nabla^2 \mathcal{L}(\boldsymbol{L}^*, \boldsymbol{R}^*)) \leq \max_{\substack{\boldsymbol{u}, \boldsymbol{v} \\ \|\boldsymbol{u}\|_2^2 + \|\boldsymbol{v}\|_2^2 = 1}} 2\big(\sigma_{\max}(\boldsymbol{L}^*)\|\boldsymbol{u}\|_2 + \sigma_{\max}(\boldsymbol{R}^*)\|\boldsymbol{v}\|_2\big)^2. \tag{87}$$

Then, by using Cauchy-Schwarz inequality,

$$\lambda_{\max}(\nabla^2 \mathcal{L}(\boldsymbol{L}^*, \boldsymbol{R}^*)) \leq 2\big(\sigma_{\max}(\boldsymbol{L}^*)^2 + \sigma_{\max}(\boldsymbol{R}^*)^2\big). \tag{88}$$

Now, we will show that this upper bound is achievable. Let us decompose $\boldsymbol{L}^*$ as $\boldsymbol{U}_L \boldsymbol{\Sigma}_L \boldsymbol{V}_L^\top$ by SVD, and denote by $\boldsymbol{u}_L$ and $\boldsymbol{v}_L$ the left and right singular vectors corresponding to the largest singular value, respectively. Moreover, decompose $\boldsymbol{R}^*$ as $\boldsymbol{U}_R \boldsymbol{\Sigma}_R \boldsymbol{V}_R^\top$ by SVD, and denote by $\boldsymbol{u}_R$ and $\boldsymbol{v}_R$ the left and right singular vectors corresponding to the largest singular value, respectively. We determine a particular $(\boldsymbol{U}^*, \boldsymbol{V}^*)$ such that it achieves the upper bound while satisfying the constraint $\|\boldsymbol{U}^*\|_F^2 + \|\boldsymbol{V}^*\|_F^2 = 1$. Choose

$$\boldsymbol{U}^{*\top} = \frac{\sigma_{\max}(\boldsymbol{L}^*)}{\sqrt{\sigma_{\max}(\boldsymbol{L}^*)^2 + \sigma_{\max}(\boldsymbol{R}^*)^2}} \boldsymbol{v}_L \boldsymbol{u}_R^\top, \quad \boldsymbol{V}^* = \frac{\sigma_{\max}(\boldsymbol{R}^*)}{\sqrt{\sigma_{\max}(\boldsymbol{L}^*)^2 + \sigma_{\max}(\boldsymbol{R}^*)^2}} \boldsymbol{u}_L \boldsymbol{v}_R^\top. \tag{89}$$

Using the fact that, for any vectors $\boldsymbol{x}$ and $\boldsymbol{y}$ $\big\|\boldsymbol{x}\boldsymbol{y}^\top\big\|_F^2 = \|\boldsymbol{x}\|_2^2 \|\boldsymbol{y}\|_2^2$,

$$2\left\|\frac{(\sigma_{\max}(\boldsymbol{L}^*)^2 + \sigma_{\max}(\boldsymbol{R}^*)^2)}{\sqrt{\sigma_{\max}(\boldsymbol{L}^*)^2 + \sigma_{\max}(\boldsymbol{R}^*)^2}} \boldsymbol{u}_L \boldsymbol{u}_R^\top\right\|_F^2 = 2(\sigma_{\max}(\boldsymbol{L}^*)^2 + \sigma_{\max}(\boldsymbol{R}^*)^2). \tag{90}$$

$\square$

# D Additional Experimental Results

## D.1 Visualization of the Contour Map of the Loss Landscape

To study the dynamics of deep matrix factorization, we analyze the trajectories of GD. Previous works have visualized neural network loss landscapes to explore their highly non-convex and non-Euclidean structure (Goodfellow et al., 2014; Li et al., 2018). However, the high-dimensionality prevents full visualization. As a result, only 1-D (line) or 2-D (surface) visualizations are available. In this paper, we focus on contour maps of the loss landscape in the vicinity of a global minimum and a methodology employed in prior studies to generate them.

**Contour Plots with Random Projections.** We want to visualize the loss landscape around a global minimum $\boldsymbol{w}^* \in \mathbb{R}^N$. We select two random vectors, $\boldsymbol{\zeta}$ and $\boldsymbol{\gamma}$, from $\mathbb{R}^N$. Then, for any $K \subset \mathbb{R}^2$, we can define the function $p : K \to \mathbb{R}$ :

$$p(x,y) = \mathcal{L}(\boldsymbol{w}^* + x\boldsymbol{\zeta} + y\boldsymbol{\gamma}), \quad \forall (x,y) \in K, \tag{91}$$

and plot $p$ with the desired resolution.

**Scale Invariance and Manifolds**. Note that our loss function is *scale-invariant*, which means that for any nonzero scalar $c \in \mathbb{R}$, multiplying one layer by $c$ and the next layer by $1/c$, or vice versa, yields the same end-to-end function. This phenomenon forms a manifold for global minimizers in the loss landscape (Dinh et al., 2017). Furthermore, we know that $\nabla^2 \mathcal{L}(\boldsymbol{w}^*)$ is rank-deficient by at least the order of $1 - 1/L$ ; that is, at least $1 - 1/L$ of the eigenvalues values of $\nabla^2 \mathcal{L}(\boldsymbol{w}^*)$ are zero (Mulayoff & Michaeli, 2020). This means that the ratio of the manifold dimension to the ambient space dimension increases as $L$ grows.

**Projection onto the Hessian Eigenvectors.** If we use random projections in visualizations, plots might not be informative to track the optimization dynamics of GD due to the phenomenon caused by the scale invariance. To make contour maps as informative as possible, we choose $\boldsymbol{\zeta}$ and $\boldsymbol{\gamma}$ to be $\boldsymbol{v}_1$ and $\boldsymbol{v}_N$, respectively — the eigenvectors of the largest and smallest eigenvalues of $\nabla^2 \mathcal{L}(\boldsymbol{w}^*)$.

### D.2  EXPERIMENT DETAILS AND ADDITIONAL EXPERIMENTS

For the experiment, we first generate the layer dimensions randomly and then construct the optimal layers $[\boldsymbol{W}_1^*, \boldsymbol{W}_2^*, \ldots, \boldsymbol{W}_L^*]$ as Gaussian random matrices, with each entry sampled from $N(0,1)$ according to the generated dimensions. Then, we compute $\boldsymbol{M}$ or $m$ by $\prod_{j=1}^L \boldsymbol{W}_j^*$. We then perform the same experiments as in Figs. 1–2, varying the depth, dimensions, and initialization distance $r$ (as shown in Figs. 3-7). We note that oscillations occur along the eigenvector corresponding to the maximum eigenvalue of the Hessian. The dimensions of the factors, i.e., $d_0, d_1, \ldots, d_L$, in Fig. 6 are given by $1, 9, 4, 8, 24, 16, 17, 11, 21, 3, 22, 3, 3, 15, 3, 18, 17, 16, 5, 12, 1$, which implies $N = 2421$, while the dimensions of the factors in Fig. 7 are given by $1, 9, 4, 8, 1$, which implies $N = 293$.

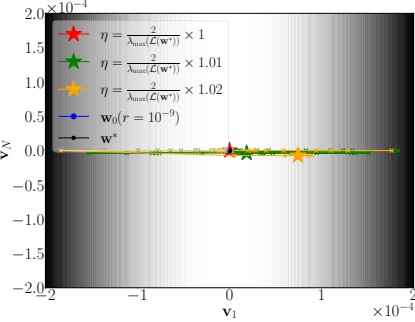

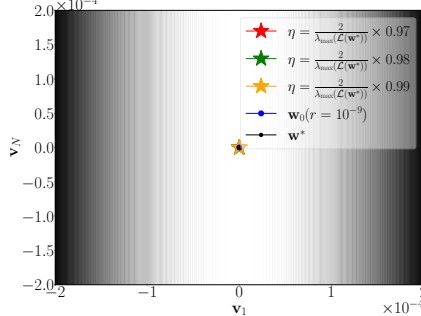

(a) Trajectories of GD initialized at $\boldsymbol{w}_0$ with step sizes $\geq 2/\lambda_{\max}$ are depicted by colored lines, and their corresponding convergence points are marked by colored $\star$ symbols.

(b) Trajectories of GD initialized at $\boldsymbol{w}_0$ with step sizes $< 2/\lambda_{\max}$ are depicted by colored lines, and their corresponding convergence points are marked by colored $\star$ symbols.

Figure 3: Contour map of the loss landscape around a minimum in a 15-layer overparameterized scalar factorization of a random scalar. GD with different step sizes $\eta$, indicated by different colors, is initialized within a radius of $10^{-9}$ from the minimum, in the direction of the Hessian eigenvector corresponding to the largest eigenvalue. The vector $\boldsymbol{v}_1$ denotes the eigenvector of the Hessian corresponding to the largest eigenvalue, while $\boldsymbol{v}_N$ denotes the eigenvector corresponding to the smallest eigenvalue. The value of $\lambda_{\max}(\nabla^2 \mathcal{L}(\boldsymbol{w}^*))$ is computed using the closed-form expression derived in Theorem 4.

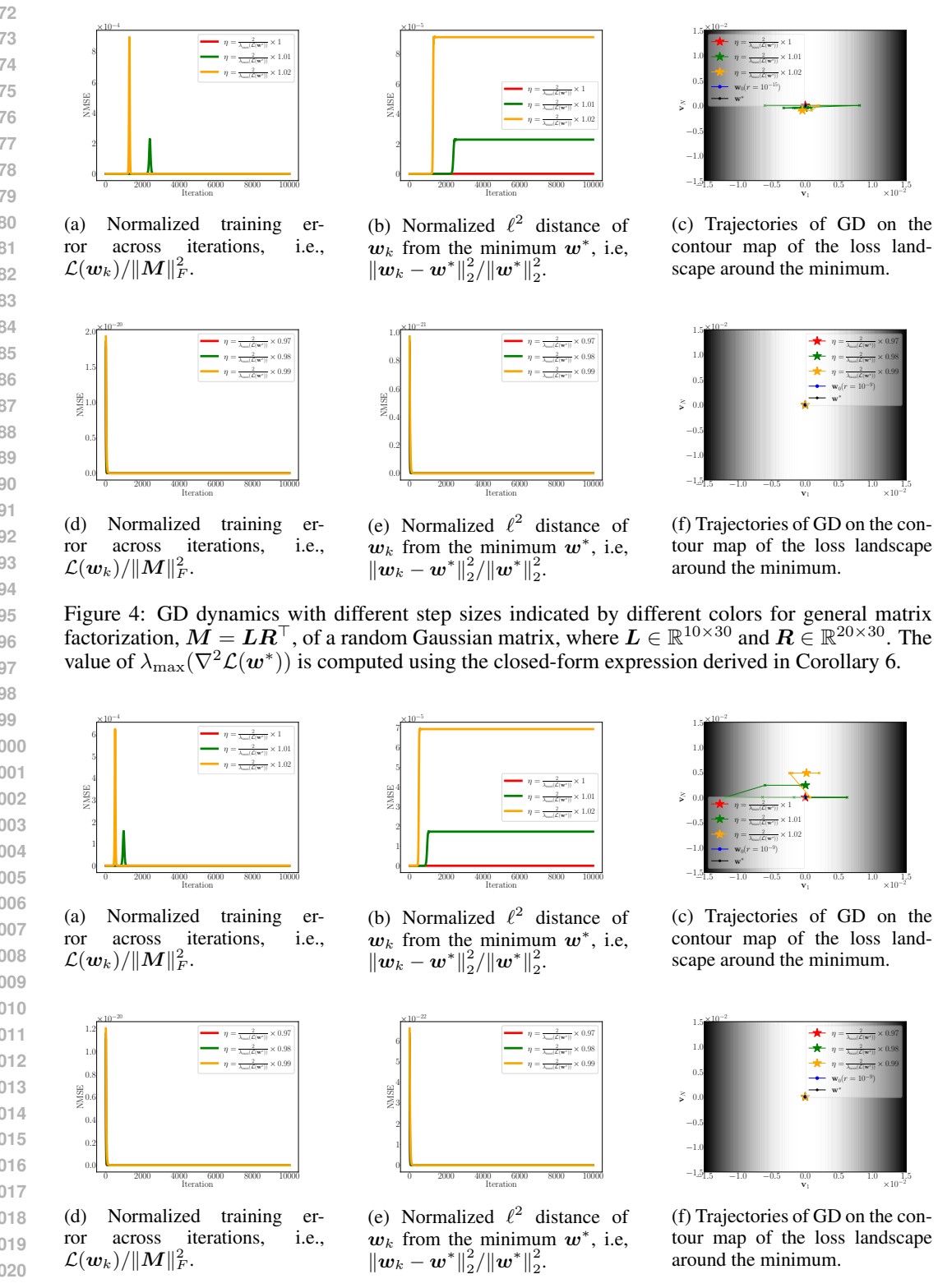

(a) Normalized training error across iterations, i.e., $\mathcal{L}(\boldsymbol{w}_k)/\|\boldsymbol{M}\|_F^2$.

(b) Normalized $\ell^2$ distance of $\boldsymbol{w}_k$ from the minimum $\boldsymbol{w}^*$, i.e, $\|\boldsymbol{w}_k - \boldsymbol{w}^*\|_2^2/\|\boldsymbol{w}^*\|_2^2$.

(c) Trajectories of GD on the contour map of the loss landscape around the minimum.

(d) Normalized training error across iterations, i.e., $\mathcal{L}(\boldsymbol{w}_k)/\|\boldsymbol{M}\|_F^2$.

(e) Normalized $\ell^2$ distance of $\boldsymbol{w}_k$ from the minimum $\boldsymbol{w}^*$, i.e, $\|\boldsymbol{w}_k - \boldsymbol{w}^*\|_2^2/\|\boldsymbol{w}^*\|_2^2$.

(f) Trajectories of GD on the contour map of the loss landscape around the minimum.

Figure 4: GD dynamics with different step sizes indicated by different colors for general matrix factorization, $\boldsymbol{M} = \boldsymbol{L}\boldsymbol{R}^\top$, of a random Gaussian matrix, where $\boldsymbol{L} \in \mathbb{R}^{10\times30}$ and $\boldsymbol{R} \in \mathbb{R}^{20\times30}$. The value of $\lambda_{\max}(\nabla^2\mathcal{L}(\boldsymbol{w}^*))$ is computed using the closed-form expression derived in Corollary 6.

(a) Normalized training error across iterations, i.e., $\mathcal{L}(\boldsymbol{w}_k)/\|\boldsymbol{M}\|_F^2$.

(b) Normalized $\ell^2$ distance of $\boldsymbol{w}_k$ from the minimum $\boldsymbol{w}^*$, i.e, $\|\boldsymbol{w}_k - \boldsymbol{w}^*\|_2^2/\|\boldsymbol{w}^*\|_2^2$.

(c) Trajectories of GD on the contour map of the loss landscape around the minimum.

(d) Normalized training error across iterations, i.e., $\mathcal{L}(\boldsymbol{w}_k)/\|\boldsymbol{M}\|_F^2$.

(e) Normalized $\ell^2$ distance of $\boldsymbol{w}_k$ from the minimum $\boldsymbol{w}^*$, i.e, $\|\boldsymbol{w}_k - \boldsymbol{w}^*\|_2^2/\|\boldsymbol{w}^*\|_2^2$.

(f) Trajectories of GD on the contour map of the loss landscape around the minimum.

Figure 5: GD dynamics with different step sizes indicated by different colors for general matrix factorization, $\boldsymbol{M} = \boldsymbol{L}\boldsymbol{R}^\top$, of a random Gaussian matrix, where $\boldsymbol{L} \in \mathbb{R}^{25\times30}$ and $\boldsymbol{R} \in \mathbb{R}^{20\times30}$. The value of $\lambda_{\max}(\nabla^2\mathcal{L}(\boldsymbol{w}^*))$ is computed using the closed-form expression derived in Corollary 6.

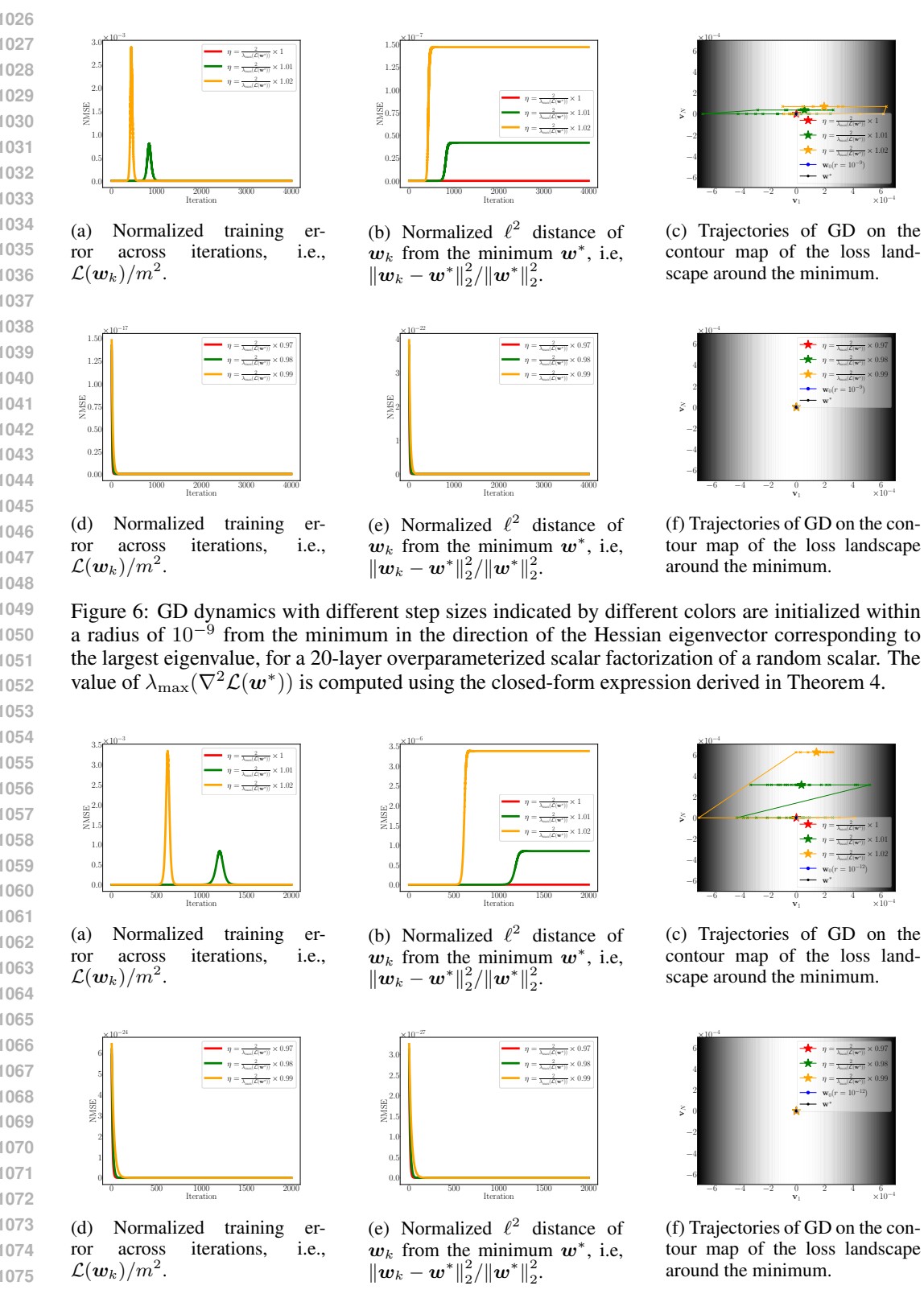

(a) Normalized training error across iterations, i.e., $\mathcal{L}(\boldsymbol{w}_k)/m^2$.

(b) Normalized $\ell^2$ distance of $\boldsymbol{w}_k$ from the minimum $\boldsymbol{w}^*$, i.e, $\|\boldsymbol{w}_k - \boldsymbol{w}^*\|_2^2/\|\boldsymbol{w}^*\|_2^2$.

(c) Trajectories of GD on the contour map of the loss landscape around the minimum.

(d) Normalized training error across iterations, i.e., $\mathcal{L}(\boldsymbol{w}_k)/m^2$.

(e) Normalized $\ell^2$ distance of $\boldsymbol{w}_k$ from the minimum $\boldsymbol{w}^*$, i.e, $\|\boldsymbol{w}_k - \boldsymbol{w}^*\|_2^2/\|\boldsymbol{w}^*\|_2^2$.

(f) Trajectories of GD on the contour map of the loss landscape around the minimum.

Figure 6: GD dynamics with different step sizes indicated by different colors are initialized within a radius of $10^{-9}$ from the minimum in the direction of the Hessian eigenvector corresponding to the largest eigenvalue, for a 20-layer overparameterized scalar factorization of a random scalar. The value of $\lambda_{\max}(\nabla^2\mathcal{L}(\boldsymbol{w}^*))$ is computed using the closed-form expression derived in Theorem 4.

(a) Normalized training error across iterations, i.e., $\mathcal{L}(\boldsymbol{w}_k)/m^2$.

(b) Normalized $\ell^2$ distance of $\boldsymbol{w}_k$ from the minimum $\boldsymbol{w}^*$, i.e, $\|\boldsymbol{w}_k - \boldsymbol{w}^*\|_2^2/\|\boldsymbol{w}^*\|_2^2$.

(c) Trajectories of GD on the contour map of the loss landscape around the minimum.

(d) Normalized training error across iterations, i.e., $\mathcal{L}(\boldsymbol{w}_k)/m^2$.

(e) Normalized $\ell^2$ distance of $\boldsymbol{w}_k$ from the minimum $\boldsymbol{w}^*$, i.e, $\|\boldsymbol{w}_k - \boldsymbol{w}^*\|_2^2/\|\boldsymbol{w}^*\|_2^2$.

(f) Trajectories of GD on the contour map of the loss landscape around the minimum.

Figure 7: GD dynamics with different step sizes indicated by different colors are initialized within a radius of $10^{-12}$ from the minimum in the direction of the Hessian eigenvector corresponding to the largest eigenvalue, for a 5-layer overparameterized scalar factorization of a random scalar. The value of $\lambda_{\max}(\nabla^2\mathcal{L}(\boldsymbol{w}^*))$ is computed using the closed-form expression derived in Theorem 4.

