# OpenReview forum: "Sharpness of Minima in Deep Matrix Factorization: Exact Expressions"
_ICLR.cc/2026/Conference — Submitted to ICLR 2026_

### Official Review · Reviewer_F5xk · 2025-10-29

**Soundness:** 3
**Presentation:** 3
**Contribution:** 2
**Rating:** 4
**Confidence:** 3

**Summary:**

In this paper, the authors study the optimization landscape of deep matrix factorization which is defined as minimizing the objective function

$$f(W_1,…,W_L) = \Vert M - W_L \cdots W_1 \Vert_F^2,$$

where $M \in \mathbb R^{d_L\times d_0}$ is a fixed ground-truth. The main contribution of the paper is an explicit formula for the worst-case sharpness of $f$, i.e., the operator norm of the Hessian, at an arbitrary point (W_1,…,W_L). Whereas existing work by Mulayoff & Michaeli (2020) already derived an explicit formula for the Hessian itself and a closed-form solution for its operator norm at sharpness minimizing global minima, they claimed that finding a closed-form expression of the worst-case sharpness is impossible in general. The authors of the present paper aim to refute this claim with their result.

**Strengths:**

+ Clearly written paper
+ Rigorous result which might be useful for the study of generalization on deep matrix factorization

**Weaknesses:**

- Imprecise comparison to existing work
- Main result is „only“ computing the derivative of a specific matrix function and as such rather feels like a technical lemma for further studies than a stand-alone theoretical result

**Questions:**

On the one hand, I acknowledge that having an explicit formula for the worst-case sharpness is a useful tool for further studies of generalization and implicit regularization in the toy setting of matrix factorization (or linear neural network training). On the other hand, I do not feel that it’s a stand-alone result for a publication yet. What exactly do we learn from having this representation? How can we use it? Consequently, I lean towards not accepting the paper.

I furthermore spotted the following issues/unclear points:

- l. 067: The description of Mulayoff & Michaeli (2020) is imprecise. In fact, they consider a slightly more general objective function in which input/output data of the linear network is included. Although they only consider full-rank data which means that the sets of global minimizers of both problems are equivalent, the data has still impact on the sharpness calculation.

- Definition 1: For which $x_0$ (2) needs to hold?

- l. 097: „… for a minimum $x_\star$…“

- l. 114: „… flatness/sharpness is measured…“

- l. 121: Maybe I missed it, but is it possible that $\lambda_\max$ without argument has not been defined?

- l. 221: To be precise, M does not denote the optimal parameters, but the product that any choice of optimal parameters will produce.

- Eq. (22)-(23): It would help to add the dimensions of U_i and u_i in the optimization problem or at least in the explanation afterwards that u_i is the vectorization of U_i.

- Eq. (29): Shouldn’t the non-negativity constraint on $x$ appear in the Lagrangian?

- l. 316: „… on the hypersphere…“

---

> ### Author Response · Authors · 2025-11-15
> **Response to Reviewer F5xk**
>
> > *On the other hand, I do not feel that it’s a stand-alone result for a publication yet. What exactly do we learn from having this representation? How can we use it?*
>
> **Response:**  The largest Hessian eigenvalue, or worst-case sharpness, is a key metric for measuring the sharpness of the loss landscape. To the best of our knowledge, [1] is the only work that derives a full characterization of the Hessian spectrum at an arbitrary point in parameter space. However, their characterization is valid only for one-hidden-layer scalar linear and ReLU networks, which are extremely simplified problem settings. Therefore, we resolved a nontrivial open question posed in [2]. Furthermore, **we extended our results to local minima** (also see *Main Revisions*). We sincerely believe that our results are a key step to understand the optimization dynamics of deep learning, which could, in turn, inform the design of more effective optimizers or regularization techniques. Additionally, there are numerous notable works that investigated the geometry of the loss landscape near a minimum [2,3,4,5].
>
>
> **Questions**
>
> > **Q1)** l. 067: The description of Mulayoff & Michaeli (2020) is imprecise. In fact, they consider a slightly more general objective function in which input/output data of the linear network is included. Although they only consider full-rank data which means that the sets of global minimizers of both problems are equivalent, the data has still impact on the sharpness calculation.
>
> > **Answer:**  Thank you for your comment. To be more precise, the maximum eigenvalue of the Hessian is defined in [2] as $\max_{||B||_F^2 = 1} 2 \Sigma _{k = 1}^{L} || ( \prod _{i = k+1}^{L} \mathbf{W}_i)^\top \mathbf{B} \mathbf{\hat{\Sigma}}_x ( \prod _{j = 1}^{k-1} \mathbf{W}_j)^\top|| _F^2 $. When $\mathbf{\Sigma_x} = \mathbf{I}$, the sharpness does not contain any geometrical or statistical information about the data. Moreover, this is exactly the same formulation that appears in the deep matrix factorization setting (see (21) in the revised manuscript). To clarify our statement, we revised the introduction (see l. 075).
>
> > **Q2)** Definition 1: For which $x_0$ (2) needs to hold?
>
> > **Answer:** $x_0$ represents the initial deviation from $x_*$, quantifying how far the algorithm starts from the optimum. Definition 1 indicates that if the dynamics begin with a small perturbation $x_0$ but this deviation grows over time, then  $x_*$ is dynamically unstable for learning rate $\eta$. This definition is simply the negation of the dynamical stability definition in [3]. To clarify this point, we revised the Definition 1 by incorporating the following sentence: ``This means that even if the dynamics begin with a small perturbation $x_0$ of $x_*$ , if this deviation grows over time, then $x_*$ is dynamically unstable for learning rate $\eta$."
>
> > **Q3)** l. 097: „… for a minimum…“
>
> > **Answer:** Thanks for your comment. We fixed it in the revised manuscript.
>
> > **Q4)** l. 114: „… flatness/sharpness is measured…“
>
> > **Answer:** We have reworded the sentence to clarify the ambiguity.
>
> > **Q5)** l. 121: Maybe I missed it, but is it possible that $\lambda_{\max}$ without argument has not been defined?
>
> > **Answer:** Thanks for your comment. We fixed it in the revised manuscript.
>
> > **Q6)** l. 221: To be precise, M does not denote the optimal parameters, but the product that any choice of optimal parameters will produce.
>
> > **Answer:** M denotes the matrix contains the parameters subject to factorization. Therefore, we revised the manuscript according to this definition. Thanks for your comment.
>
> > **Q7)** Eq. (22)-(23): It would help to add the dimensions of U_i and u_i in the optimization problem or at least in the explanation afterwards that u_i is the vectorization of U_i.
>
> > **Answer:**  Thanks for your concern. We updated the manuscript.
>
> > **Q8)** Eq. (29): Shouldn’t the non-negativity constraint on $x$ appear in the Lagrangian?
>
> > **Answer:** We have greatly shortened this argument via a direct application of Cauchy–Schwarz in the revised manuscript, so we no longer consider the Lagrangian.
>
> > **Q9)** l. 316: „… on the hypersphere…“
>
> > **Answer:** Thanks for your comment. We wanted to use the term hypersphere to imply that $\sum_{i=1}^L ||U_i||_F^2 = 1$. However, we agree that it might be confusing. Therefore, we omitted the term "hypersphere".
>
> [1] S. P. Singh and T. Hofmann, “Closed form of the hessian spectrum for some neural networks, 2024.
>
> [2] R. Mulayoff and T. Michaeli, “Unique properties of flat minima in deep networks,” in ICML, PMLR, 2020.
>
> [3] L. Wu, C. Ma, et al., “How sgd selects the global minima in overparameterized learning: A dynamical stability perspective,” NeurIPS, 2018.
>
> [4] C. Josz, On the geometry of flat minima, 2025. arXiv: 2509.11386.
>
> [5] "Learning Dynamics of Deep Linear Networks Beyond the Edge of Stability", Avrajit Ghosh, Soo Min Kwon, Rongrong Wang, Saiprasad Ravishankar, Qing Qu. ICLR 2025.

---

> ### Author Response · Authors · 2025-11-17
> **What exactly we do learn from Theorem 5:  An Important Corollary!**
>
> We would like to mention a subtle consequence of Theorem 5. Consider a depth-2 matrix factorization problem, i.e.,
> $$
> \min_{\mathbf{L},\mathbf{R}} \mathcal{L}(\mathbf{L},\mathbf{R}), \quad \text{such that} \quad  \mathcal{L}(\mathbf{L},\mathbf{R}) = ||\mathbf{M} - \mathbf{L}\mathbf{R}^\top||_F^2,
> $$
> where $\mathbf{M} \in \mathbb{R}^{m \times n}, \mathbf{L} \in \mathbb{R}^{m \times k}$ and $\mathbf{R} \in \mathbb{R}^{n \times k}$. Suppose $m=n=k=3$ and $\mathbf{M} = \text{diag}(11,2,1)$. We know that for any flat minimum $(\mathbf{L'},\mathbf{R'})$, $\lambda _\max(\nabla ^2(\mathcal{L}(\mathbf{L'},\mathbf{R'}))) = 4 \times \sigma _\max (M)$, which equals to $44$ [1].  By definition, any minimum whose sharpness is equal to $44$ is a flat minimum. Let's investigate a specific minimizer $(\mathbf{L} ^{*}, \mathbf{R} ^ * )$ such that $\mathbf{L} ^ * = \text{diag}(\sqrt{11} , \frac{2}{3}, 1)$ and $\mathbf{R} ^ * = \text{diag}(\sqrt{11} , 3, 1)$. By using Corollary $6$, $\lambda _\max(\nabla ^2(\mathcal{L}(\mathbf{L} ^ *,\mathbf{R} ^ *))) = 44$, which means that $(\mathbf{L} ^ *,\mathbf{R} ^ *)$ is a flat minimum.  On the other hand, by definition of balancedness, i.e., $\mathbf{L}\mathbf{L}^\top = \mathbf{R}^\top \mathbf{R}$, $(\mathbf{L} ^ *, \mathbf{R} ^ *)$ is not a balanced minimizer. This means that any flat minimum is not necessarily a balanced minimizer. **This observation is a direct consequence of Theorem 5 of our paper.** This observation refutes the result of [2] that has shown that a balanced minimum is also flat.
>
> [1] R. Mulayoff and T. Michaeli, “Unique properties of flat minima in deep networks,” in ICML, PMLR, 2020.
>
> [2] "Learning Dynamics of Deep Linear Networks Beyond the Edge of Stability", Avrajit Ghosh, Soo Min Kwon, Rongrong Wang, Saiprasad Ravishankar, Qing Qu. ICLR 2025.

---

> ### Comment · Reviewer_F5xk · 2025-11-21
>
> I thank the authors for their replies to my comments. Here some further remarks:
>
> Regarding Q1: You are right, when screening the paper I missed their assumption A.2 which they require to calculate the operator norm of the Hessian. Sorry for this mistake.
>
> Regarding Q2: Let me clarify my question. My comment shall mainly point out that Definition 1 is imprecise at the moment. In fact, one has to clarify whether (2) has to hold for all x_0 in some eps-neighborhood of x_* or just for at least one specific initialization. (Right now it reads as if the second were true, which wouldn't make sense to me). I would have the same comment on the original definition in Wu et al.
>
> Regarding "Furthermore, we extended our results to local minima (also see Main Revisions).": In my opinion, this is overstated. If I read your comment in Section 4 correctly, you just observe that by results of others there are no local minima that are not global in the problem setting you consider. This is a useful observation, but no extension of the results.
>
> Regarding the observation on a relation between flat and balanced minima in shallow matrix factorization: This is an interesting insight. Especially in light of [1], which shows that balanced minimizers and sharpness minimizers (measured in terms of scaled trace of the Hessian) agree. However, I still think that various applications of this type should be discussed at greater length in your paper (working out how they relate) to make it ripe for publication.
>
> [1] Dingh et. al; "Flat minima generalize for low-rank matrix recovery"

---

> ### Author Response · Authors · 2025-11-28
> **Response to Reviewer F5xk**
>
> Thank you for the clarification. We have substantially revised Definition 1 to make it as clear as possible. Note that the dynamically unstable minimum definition relies on how valid the quadratic approximation of the loss landscape near minimum is.
>
> According to Definition 1, the residuals follow the dynamics given below.
>
> $$
>   \epsilon _{t+1} =  (I - \eta \nabla ^2 \mathcal{L}(w _*)) ^{t+1} \epsilon_0.
> $$
>
> This means that if the absolute value of every eigenvalue of $(I - \eta \nabla ^2 \mathcal{L}(w_*))$ is less than 1 then
> $\lim_{t \rightarrow {\infty}} \epsilon_t  = 0 $, i.e., almost sure convergence or stable convergence. However, if any eigenvalue of $(I - \eta \nabla ^2 \mathcal{L}(w_*))$ is larger than 1, i.e., there exists an index $k \in [N]$ such that $\lambda_k > 2/\eta$, whether divergence occurs depends on the subspace where the perturbation lies. To summarize, if $\epsilon_0$ is orthogonal to the eigenspace corresponding to the eigenvalues that are larger than $2/\eta$, then GD converges even though the minimum is dynamically unstable. This means that even though a minimum is dynamically unstable, there exists a manifold in which the perturbation lies such that GD converges. To be clear, the convergence meant here is not almost sure convergence. It is $ \lim_{t \rightarrow \infty}\mathcal{L}(w_t) = \mathcal{L}(w_*)$ where $w _*$ is any minimizer of the loss.  We explained this phenomenon in detail in Section 7.
>
> > *Regarding "Furthermore, we extended our results to local minima (also see Main Revisions).": In my opinion, this is overstated. If I read your comment in Section 4 correctly, you just observe that by results of others there are no local minima that are not global in the problem setting you consider. This is a useful observation, but no extension of the results.*
>
> **Response:** We agree with you. This statement might be an overstatement. Nonetheless, here, we meant to say that overparameterized deep matrix factorization problems with convex, differentiable loss functions do not have spurious minima. We have moved the paragraph that mentions this observation and now state in Section 2, where we defined our problem, that our problem does not have spurious minima. Thank you for your observation.
>
> > *Regarding the observation on a relation between flat and balanced minima in shallow matrix factorization: This is an interesting insight. Especially in light of [1], which shows that balanced minimizers and sharpness minimizers (measured in terms of scaled trace of the Hessian) agree. However, I still think that various applications of this type should be discussed at greater length in your paper (working out how they relate) to make it ripe for publication.*
>
> **Response:** Thank you for your valuable insights. We have substantially revised the paper. The revisions are as follows:
>
> We added Section 5, **a new section that reveals remarkable aspects of flat minima** in general deep matrix factorization problems. These findings, which are **direct consequences of our Theorem 5**, are as follows:
>
> - A minimizer of deep overparameterized scalar factorization loss is flat **if and only if** the product of spectral norms of left and right intermediate factors is constant across layers *(Corollary 7)*.
>
> - Flat minima are spectral-norm balanced in depth-2 matrix factorization *(Corollary 8)*. This implies that **flat minima are not necessarily Frobenius-norm balanced, contrary to claims made in several works [1,2]**.
>
> - A minimizer of deep matrix factorization loss is flat **if** the product of spectral norms of left and right intermediate factors is constant across layers *(Corollary 10)*.
>
> We have added **a Discussion section, which was absent in the previous draft**. In this section, we discussed the further implications of the results presented in Section 5. We thoroughly examined claims in [1, 2] that flat minima coincide with Frobenius-norm balanced minima in depth-2 matrix factorization. We also discussed how misleading current sharpness measures can be for loss landscape analysis. We concluded this section by highlighting **the need for a new, robust sharpness measure that unifies the insights from existing ones.**
>
> Furthermore, we have revised the **entire Experiment section** to clarify issues related to $x _0$. Additionally, we moved the proof of Theorem 4 to the Appendix and revised Conclusion section.
>
> [1] L. Ding, D. Drusvyatskiy, M. Fazel, and Z. Harchaoui, “Flat minima generalize for low-rank matrix recovery,” 2024.
>
> [2] "Learning Dynamics of Deep Linear Networks Beyond the Edge of Stability", Avrajit Ghosh, Soo Min Kwon, Rongrong Wang, Saiprasad Ravishankar, Qing Qu. ICLR 2025.

---

### Official Review · Reviewer_BgPX · 2025-10-30

**Soundness:** 3
**Presentation:** 3
**Contribution:** 1
**Rating:** 2
**Confidence:** 4

**Summary:**

The main contribution of this paper is the derivation of the maximum eigenvalue of the Hessian of the squared-error loss at any minimizer in general overparameterized deep matrix factorization. The authors refute the claim that deriving a closed-form expression for arbitrary global minima is intractable, which they demonstrate in Theorem 5. This contrasts with prior work, which characterizes the Hessian eigenvalues only at balanced minima (e.g., [1]), a setting that is significantly simpler than the general case. In addition, the authors empirically show that gradient descent consistently escapes dynamically unstable minima, regardless of how close the initialization is to such a point.

[1] "Learning Dynamics of Deep Linear Networks Beyond the Edge of Stability", Avrajit Ghosh, Soo Min Kwon, Rongrong Wang, Saiprasad Ravishankar, Qing Qu. ICLR 2025.

**Strengths:**

I find the paper well written and generally easy to follow. Since the authors derive a general closed-form expression for the maximum eigenvalue of the Hessian, something that to my knowledge has not been shown before, the paper does make a theoretical contribution. However, I do not believe that this derivation alone is sufficient to justify acceptance. The additional empirical claim that “GD always escapes from a dynamically unstable minimum, regardless of how close the initialization is to that minimum” has already been demonstrated in Ghosh et al. (2025). I discuss this further in the weaknesses section.

**Weaknesses:**

As discussed previously, I believe the empirical observation has already been made by Ghosh et al. (2025). Their contour plots demonstrate that, when initialized at an arbitrary unbalanced point, gradient descent transitions away from dynamically unstable minima toward the flattest minimum (i.e., the balanced solution). The flattest minimum is sufficient to induce periodic oscillations, whereas the other minima, being sharper, cannot sustain them. If I am misunderstanding the distinction, could the authors clarify how their experiments differ from those presented by Ghosh et al. (2025)? Without a clear differentiation, I am concerned that the empirical component does not provide additional novelty.

Given this, it appears that the main contribution of the paper is Theorem 5. While the derivation is interesting, I am not convinced that it is sufficient on its own to justify acceptance. One possibility is that the authors could use the result from Theorem 5 to theoretically demonstrate why unstable minima cannot sustain oscillations, which would strengthen the empirical story and align it more closely with the observations from Ghosh et al. (2025). However, even with such an extension, I am uncertain whether this would constitute a sufficiently substantial contribution.

**Questions:**

I have asked a few questions in the weaknesses section.

---

> ### Author Response · Authors · 2025-11-17
> **Response to Reviewer BgPX**
>
> We would like to thank you for your valuable insights.
>
> > *As discussed previously, I believe the empirical observation has already been made by Ghosh et al. (2025). Their contour plots demonstrate that, when initialized at an arbitrary unbalanced point, gradient descent transitions away from dynamically unstable minima toward the flattest minimum (i.e., the balanced solution).*
>
> First of all, we want to clarify a point in your comment quoted above. Consider a depth-2 matrix factorization problem, i.e.,
> $$
> \min_{\mathbf{L},\mathbf{R}} \mathcal{L}(\mathbf{L},\mathbf{R}), \quad \text{such that} \quad  \mathcal{L}(\mathbf{L},\mathbf{R}) = ||\mathbf{M} - \mathbf{L}\mathbf{R}^\top||_F^2,
> $$
> where $\mathbf{M} \in \mathbb{R}^{m \times n}, \mathbf{L} \in \mathbb{R}^{m \times k}$ and $\mathbf{R} \in \mathbb{R}^{n \times k}$. Suppose $m=n=k=3$ and $\mathbf{M} = \text{diag}(11,2,1)$. We know that for any flat minimum $(\mathbf{L'},\mathbf{R'})$, $\lambda _\max(\nabla ^2(\mathcal{L}(\mathbf{L'},\mathbf{R'}))) = 4 \times \sigma _\max (M)$, which equals to $44$ [1].  By definition, any minimum whose sharpness is equal to $44$ is a flat minimum. Let's investigate a specific minimizer $(\mathbf{L} ^{*}, \mathbf{R} ^ * )$ such that $\mathbf{L} ^ * = \text{diag}(\sqrt{11} , \frac{2}{3}, 1)$ and $\mathbf{R} ^ * = \text{diag}(\sqrt{11} , 3, 1)$. By using Corollary $6$, $\lambda _\max(\nabla ^2(\mathcal{L}(\mathbf{L} ^ *,\mathbf{R} ^ *))) = 44$, which means that $(\mathbf{L} ^ *,\mathbf{R} ^ *)$ is a flat minimum.  On the other hand, by definition of balancedness, i.e., $\mathbf{L}\mathbf{L}^\top = \mathbf{R}^\top \mathbf{R}$, $(\mathbf{L} ^ *, \mathbf{R} ^ *)$ is not a balanced minimizer. This means that any flat minimum is not necessarily a balanced minimizer. **This observation is a direct consequence of Theorem 5 of our paper**, and it clearly contradicts the statement in Ghosh et al. (2025): *"We further showed that balanced minima correspond to the flattest minima"*. Additionally, this specific $\mathbf{M}$ selection, fits the assumptions made in Section 2 of Ghosh et al. (2025).
>
> According to the experimental details of Ghosh et al. (2025) (see B.1 in Ghosh et al. (2025)), the considered experimental setting is a two-layer scalar network, which is an extremely simplified toy example. Having an exact expression for the maximum eigenvalue of the Hessian of the loss at arbitrary minima gives great flexibility to run these experiments in a deep matrix factorization setting. Furthermore, our claim was not about re-exploring the escape phenomenon. This phenomenon has been observed in numerous works published prior to Ghosh et al. (2025). The most notable one is [2]. Our intention was to demonstrate the escape phenomenon within overparameterized deep matrix factorization in a setting that crucially relies on our exact expression for the sharpness.
>
> [1] R. Mulayoff and T. Michaeli, “Unique properties of flat minima in deep networks,” in ICML, PMLR, 2020.
>
> [2] L. Wu, C. Ma, et al., “How sgd selects the global minima in overparameterized learning: A dynamical stability perspective,” NeurIPS, 2018.

---

> ### Author Response · Authors · 2025-11-28
> **Response to Reviewer BgPX**
>
> We have substantially revised the paper. The revisions are as follows:
>
> We added Section 5, **a new section that reveals remarkable aspects of flat minima** in general deep matrix factorization problems. These findings, which are **direct consequences of our Theorem 5**, are as follows:
>
> - A minimizer of deep overparameterized scalar factorization loss is flat **if and only if** the product of spectral norms of left and right intermediate factors is constant across layers *(Corollary 7)*.
>
> - Flat minima are spectral-norm balanced in depth-2 matrix factorization *(Corollary 8)*. This implies that **flat minima are not necessarily Frobenius-norm balanced, contrary to claims made in several works [1,2]**.
>
> - A minimizer of deep matrix factorization loss is flat **if** the product of spectral norms of left and right intermediate factors is constant across layers *(Corollary 10)*.
>
> We have added **a Discussion section, which was absent in the previous draft**. In this section, we discussed the further implications of the results presented in Section 5. We thoroughly examined claims in [1, 2] that flat minima coincide with Frobenius-norm balanced minima in depth-2 matrix factorization. We also discussed how misleading current sharpness measures can be for loss landscape analysis. We concluded this section by highlighting **the need for a new, robust sharpness measure that unifies the insights from existing ones.**
>
> Furthermore, we have revised the **entire Experiment section** to clarify issues related to $x _0$. Additionally, we moved the proof of Theorem 4 to the Appendix and revised Conclusion section.
>
> [1] L. Ding, D. Drusvyatskiy, M. Fazel, and Z. Harchaoui, “Flat minima generalize for low-rank matrix recovery,” 2024.
>
> [2] "Learning Dynamics of Deep Linear Networks Beyond the Edge of Stability", Avrajit Ghosh, Soo Min Kwon, Rongrong Wang, Saiprasad Ravishankar, Qing Qu. ICLR 2025.

---

### Official Review · Reviewer_aQPP · 2025-10-30

**Soundness:** 3
**Presentation:** 3
**Contribution:** 2
**Rating:** 2
**Confidence:** 2

**Summary:**

This paper derive the explicit expression for the eigenvalues of Hessian of the deep matrix factorization problem. There are also experiments showing the behavior of GD when the learning rate is around the stability limit $2/\lambda_{\max}(Hessian)$.

**Strengths:**

The largest eigenvalue of Hessian is a important concept in machine learning, and the authors derive the explicit expression of it although only for deep matrix factorization. This still provides some insights.

**Weaknesses:**

The main concern is the results of this paper may not be enough for a conference paper. The main theoretical result is the derivation of the largest eigenvalue of Hessian. For the other claim, 'GD escaping from unstable minima' is more or less a known fact in dynamical systems, and there are also stable manifold if the authors do not specify the initial condition $x_0$. The experiments do not seem to introduce new things.

**Questions:**

- Definition 1: Please specify the requirement or the dependency of learning rate $\eta$, as well as $x_0$. The definition itself is not complete.
- The authors claim that GD always escape from dynamically unstable minimum. However, even for unstable minima, there could still be stable manifold, although this is a lower dimensional manifold. This goes back to the problem of not specifying initial condition $x_0$.

---

> ### Author Response · Authors · 2025-11-17
> **Response to Reviewer aQPP**
>
> > *The main concern is the results of this paper may not be enough for a conference paper. The main theoretical result is the derivation of the largest eigenvalue of Hessian.*
>
> **Response:** The largest Hessian eigenvalue, or worst-case sharpness, is a key metric for measuring the sharpness of the loss landscape. To the best of our knowledge, [1] is the only work that derives a full characterization of the Hessian spectrum at an arbitrary point in parameter space. However, their characterization is valid only for one-hidden-layer scalar linear and ReLU networks, which are extremely simplified problem settings. Therefore, we resolved a nontrivial open question posed in [2]. Furthermore, we extended our results to local minima (also see Main Revisions). We sincerely believe that our results are a key step to understand the optimization dynamics of deep learning, which could, in turn, inform the design of more effective optimizers or regularization techniques. Additionally, there are numerous notable works that investigated the geometry of the loss landscape near a minimum [2,3,4].
>
> > *For the other claim, 'GD escaping from unstable minima' is more or less a known fact in dynamical systems. The experiments do not seem to introduce new things.*
>
> **Response:** We agree with you about your concern. However, our claim was not about re-exploring the escape phenomenon. Our intention was to demonstrate the escape phenomenon within overparameterized deep matrix factorization in a setting that crucially relies on our exact expression for the sharpness.
>
> **Questions:**
>
> > **Q1)** Definition 1: Please specify the requirement or the dependency of learning rate $\eta$, as well as $x_0$. The definition itself is not complete.
>
> > **Answer:** $x_0$ represents the initial deviation from $x_*$, quantifying how far the algorithm starts from the optimum. Definition 1 indicates that if the dynamics begin with a small perturbation $x_0$ but this deviation grows over time, then  $x_*$ is dynamically unstable for learning rate $\eta$. This definition is simply the negation of the dynamical stability definition in [3]. To clarify this point, we revised the Definition 1 by incorporating the following sentence: ``This means that even if the dynamics begin with a small perturbation $x_0$ of $x_*$ , if this deviation grows over time, then $x_*$ is dynamically unstable for learning rate $\eta$."
>
> > **Q2)** The authors claim that GD always escape from dynamically unstable minimum. However, even for unstable minima, there could still be stable manifold, although this is a lower dimensional manifold. This goes back to the problem of not specifying initial condition $x_0$.
>
> > **Answer:** In the experiments, to observe escape phenomenon clearly, we choose the perturbation direction for the initial point to be the eigenvector of $\nabla ^2 \mathcal{L}(\mathbf{w} ^* )$ corresponding to the largest eigenvalue, so as to avoid choosing a direction that is orthogonal to the eigenspace corresponding to the eigenvalues larger than $2/ \eta$. Note that, we could have also considered a random perturbation direction uniformly from $\mathbb{S} ^{N-1} $. [2] showed that for the deep matrix factorization problem (which has the same Hessian structure with deep linear networks), the Hessian is rank-deficient at all minima by at least the order of $1 - (1/L)$. This means that at least $1 - (1/L)$ of the eigenvalues are zero at a minimum. Thus, the probability of choosing a direction that is orthogonal to the eigenspace corresponding to eigenvalues larger than $2/  \eta$ is 0. Therefore, random perturbations would lead to the escape phenomenon with probability $1$. On the other hand, for stable minima, it is important to choose the direction as the eigenvector of $\nabla ^2 \mathcal{L}(\mathbf{w} ^* )$ corresponding to the largest eigenvalue. The reasoning is the same. If you choose a direction that is not orthogonal to the eigenspace corresponding to the eigenvalues that are zero, then GD never converges to $\mathbf{w} ^* $. This means that if we choose the perturbation direction randomly, then with probability 1, we choose a direction that is not orthogonal to the eigenspace corresponding to the eigenvalues that are zero. Therefore, for the experiment, it is convenient to choose perturbation direction as the eigenvector of $\nabla ^2 \mathcal{L}(\mathbf{w} ^* )$ corresponding to the largest eigenvalue. Since this is not trivial to see in the first sight, we updated the Section 5. Thanks to your observation.
>
> [1] S. P. Singh and T. Hofmann, “Closed form of the hessian spectrum for some neural networks, 2024.
>
> [2] R. Mulayoff and T. Michaeli, “Unique properties of flat minima in deep networks,” in ICML, PMLR, 2020.
>
> [3] L. Wu, C. Ma, et al., “How sgd selects the global minima in overparameterized learning: A dynamical stability perspective,” NeurIPS, 2018.
>
> [4] C. Josz, On the geometry of flat minima, 2025. arXiv: 2509.11386.

---

> > ### Comment · Reviewer_aQPP · 2025-11-26
> >
> > I would like to thank the authors for the response.
> >
> > For definition 1, yes, $x_0$ is a perturbation but there is no specification of the region of $x_0$. For example, there could be a manifold such that for all perturbations on that manifold (like the author said in their response to Q2), the deviation grows, while all other perturbations do not. It's unclear if your definition needs to hold for all $x_0$; if so, please specify.
> >
> > Overall, I totally understand the importance of the study of loss landscape. However, in my opinion, the contents are not enough for a conference publication. I will keep my score.

---

> ### Author Response · Authors · 2025-11-28
> **Response to Reviewer aQPP**
>
> > *For definition 1, yes, $x_0$ is a perturbation but there is no specification of the region of $x_0$. For example, there could be a manifold such that for all perturbations on that manifold (like the author said in their response to Q2), the deviation grows, while all other perturbations do not. It's unclear if your definition needs to hold for all $x_0$; if so, please specify.*
>
> **Response:** Thank you for the clarification. We have substantially revised Definition 1 to make it as clear as possible. Note that the dynamically unstable minimum definition relies on how valid the quadratic approximation of the loss landscape near minimum is.
>
> According to Definition 1, the residuals follow the dynamics given below.
>
> $$
>   \epsilon _{t+1} =  (I - \eta \nabla ^2 \mathcal{L}(w _*)) ^{t+1} \epsilon_0.
> $$
>
> This means that if the absolute value of every eigenvalue of $(I - \eta \nabla ^2 \mathcal{L}(w_*))$ is less than 1 then
> $\lim_{t \rightarrow {\infty}} \epsilon_t  = 0 $, i.e., almost sure convergence or stable convergence. However, if any eigenvalue of $(I - \eta \nabla ^2 \mathcal{L}(w_*))$ is larger than 1, i.e., there exists an index $k \in [N]$ such that $\lambda_k > 2/\eta$, whether divergence occurs depends on the subspace where the perturbation lies. To summarize, if $\epsilon_0$ is orthogonal to the eigenspace corresponding to the eigenvalues that are larger than $2/\eta$, then GD converges even though the minimum is dynamically unstable. This means that even though a minimum is dynamically unstable, there exists a manifold in which the perturbation lies such that GD converges. To be clear, the convergence meant here is not almost sure convergence. It is $ \lim_{t \rightarrow \infty}\mathcal{L}(w_t) = \mathcal{L}(w_*)$ where $w _*$ is any minimizer of the loss.  We explained this phenomenon in detail in Section 7.
>
> > *Overall, I totally understand the importance of the study of loss landscape. However, in my opinion, the contents are not enough for a conference publication. I will keep my score.*
>
>
> **Response:** In this draft, we have substantially revised the paper. The revisions are as follows:
>
> We added Section 5, **a new section that reveals remarkable aspects of flat minima** in general deep matrix factorization problems. These findings, which are **direct consequences of our Theorem 5**, are as follows:
>
> - A minimizer of deep overparameterized scalar factorization loss is flat **if and only if** the product of spectral norms of left and right intermediate factors is constant across layers *(Corollary 7)*.
>
> - Flat minima are spectral-norm balanced in depth-2 matrix factorization *(Corollary 8)*. This implies that **flat minima are not necessarily Frobenius-norm balanced, contrary to claims made in several works [1,2]**.
>
> - A minimizer of deep matrix factorization loss is flat **if** the product of spectral norms of left and right intermediate factors is constant across layers *(Corollary 10)*.
>
> We have added **a Discussion section, which was absent in the previous draft**. In this section, we discussed the further implications of the results presented in Section 5. We thoroughly examined claims in [1, 2] that flat minima coincide with Frobenius-norm balanced minima in depth-2 matrix factorization. We also discussed how misleading current sharpness measures can be for loss landscape analysis. We concluded this section by highlighting **the need for a new, robust sharpness measure that unifies the insights from existing ones.**
>
> Furthermore, we have revised the **entire Experiment section** to clarify issues related to $x _0$. Additionally, we moved the proof of Theorem 4 to the Appendix and revised Conclusion section.
>
> [1] L. Ding, D. Drusvyatskiy, M. Fazel, and Z. Harchaoui, “Flat minima generalize for low-rank matrix recovery,” 2024.
>
> [2] "Learning Dynamics of Deep Linear Networks Beyond the Edge of Stability", Avrajit Ghosh, Soo Min Kwon, Rongrong Wang, Saiprasad Ravishankar, Qing Qu. ICLR 2025.

---

### Official Review · Reviewer_z1gb · 2025-11-01

**Soundness:** 3
**Presentation:** 3
**Contribution:** 2
**Rating:** 4
**Confidence:** 3

**Summary:**

This paper studies the maximum eigenvalue of the Hessian at global minimizers of deep matrix factorization. It provides exact, closed-form expressions for both deep overparameterized scalar and matrix factorizations. Using these expressions, the authors conduct controlled gradient descent experiments near minima and observe the escape phenomenon when $\eta > 2/\lambda_\max$, consistent with results reported in prior special cases.

**Strengths:**

- Understanding the loss landscape and its geometry is an important research problem.
- This paper presents an exact formula for computing the largest eigenvalue of the Hessian at global minima in deep matrix factorization, which appears to be new.

**Weaknesses:**

- The setting considered here appears to be limited to the well-specified case (where the minimum loss is zero). Is it possible to extend the analysis to more general cases or characterize the structure of local minima?
- The experiments seems not too surprising that gradient descent escapes when the sharpness exceeds $2/\eta$.

Overall, the paper makes a valuable contribution by providing a formula for the largest eigenvalue of the Hessian at global minima. However, I believe further work may be needed before the paper is ready for publication, such as analyzing the full Hessian structure or extending the results to local minima.

**Questions:**

-	What additional insights can we obtain about the full spectrum of eigenvalues and eigenvectors Hessian or the Hessian at local minima?
-	The current derivation from Eqs. (26)/(27) to (33) seems somewhat complicated. It appears that the result could follow directly from the Cauchy–Schwarz inequality, rather than being formulated as a constrained optimization problem.
-	Typo: In Eq. (42), there is an extra period before the last $I$.

---

> ### Author Response · Authors · 2025-11-15
> **Response to Reviewer z1gb**
>
> > *The setting considered here appears to be limited to the well-specified case (where the minimum loss is zero). Is it possible to extend the analysis to more general cases...?*
>
> **Response:** We would like to start by addressing your first remark concerning the limitation to the case where the minimum loss is zero. We want to clarify this by asking two questions and answering them. First question is as follows: *why do we want to investigate the loss landscape near a minimizer, and is the minimum loss zero?*
>
> We want to investigate the loss landscape near a minimizer since overparameterized neural networks with zero training error still perform well on the test set, in contradiction to the accepted statistical wisdom on overfitting [1]. This phenomenon is usually attributed to a mechanism within gradient-based optimization methods that makes them perform so well for modern non-convex deep architectures [2]. Therefore, by this insight, we believe that it is important to investigate the behavior of gradient descent, near a minimizer of the loss function, which is strongly influenced by the geometry of loss landscape. Additionally, there are numerous notable works that investigate the sharpness of loss landscape near a minimum [3,4,5].
>
> Since deep matrix factorization problems are analogous to deep linear network training, we think that it is natural to investigate the sharpness of the loss landscape of overparameterized deep matrix factorization problems near a minimizer. Furthermore, in our case, the minimizer must have zero training error because the assumption in Equation (16) makes all points in $\mathbb{R}^{d_L \times d_0}$ feasible for factorization. Therefore, for any unregularized overparameterized model with nonnegative loss  that enables the factorization of all points in $\mathbb{R}^{d_L \times d_0}$, the minimum loss must be zero. For instance, in contrast, for a regularized deep matrix factorization problem, a minimizer does not have to have zero training error.
>
> The second question is as follows: *Can we extend this characterization for any arbitrary point in the parameter space?* Deriving a closed-form expression for the maximum eigenvalue of the Hessian of the loss function at arbitrary point in parameter space is an intractable task. To the best of our knowledge, [6] is the only work that derives a full characterization of the Hessian spectrum at an arbitrary point in parameter space. However, their characterization is valid only for one-hidden-layer scalar linear and ReLU networks, which are extremely simplified problem settings.
>
> > *Is it possible to characterize the structure of local minima?*
>
> **Response:** We would like to thank you for your comment regarding the characterization of local minima. **It is well known in the literature that overparameterized deep matrix factorization problems with convex and differentiable loss functions do not have spurious minima**; in other words, every local minimum is global. In particular, this was shown in [7].  We updated the manuscript.
>
> > *The experiments seems not too surprising that gradient descent escapes when the sharpness exceeds.*
>
> **Response:** We agree with you about your concern. However, our claim was not about re-exploring the escape phenomenon. Our intention was to demonstrate the escape phenomenon in a setting that **crucially relies on our exact expression for the sharpness.**
>
> **Questions**
>
> > **Q1)** What additional insights can we obtain about the full spectrum of eigenvalues and eigenvectors Hessian or the Hessian at local minima?
>
> > **Answer:** See response above.
>
> > **Q2)**  The current derivation from Eqs. (26)/(27) to (33) seems somewhat complicated. It appears that the result could follow directly from the Cauchy–Schwarz inequality, rather than being formulated as a constrained optimization problem.
>
> >  **Answer:** We completely agree with you. Therefore, we greatly reduced the length of the proof. Thank to your observation.
>
> > **Q3)** Typo: In Eq. (42), there is an extra period before the last $I$.
>
> >  **Answer:** We fixed the typo.
>
> [1] M. Belkin, D. Hsu, S. Ma, and S. Mandal, “Reconciling modern machine learning practice and the classical bias–variance trade-off,” Proceedings of the National Academy of Sciences, 2019.
>
> [2] B. Neyshabur, S. Bhojanapalli, D. McAllester, and N. Srebro, “Exploring generalization in deep learning,” NeurIPS, 2017.
>
> [3] L. Wu, C. Ma, et al., “How sgd selects the global minima in overparameterized learning: A dynamical stability perspective,” NeurIPS, 2018.
>
> [4] R. Mulayoff and T. Michaeli, “Unique properties of flat minima in deep networks,” in ICML, PMLR, 2020.
>
> [5] C. Josz, On the geometry of flat minima, 2025. arXiv: 2509.11386.
>
> [6] S. P. Singh and T. Hofmann, “Closed form of the hessian spectrum for some neural networks, 2024.
>
> [7] T. Laurent and J. Brecht, “Deep linear networks with arbitrary loss: All local minima are global,” in ICML, PMLR, 2018.

---

> ### Author Response · Authors · 2025-11-28
> **Response to Reviewer z1gb**
>
> We have substantially revised the paper. The revisions are as follows:
>
> We added Section 5, **a new section that reveals remarkable aspects of flat minima** in general deep matrix factorization problems. These findings, which are **direct consequences of our Theorem 5**, are as follows:
>
> - A minimizer of deep overparameterized scalar factorization loss is flat **if and only if** the product of spectral norms of left and right intermediate factors is constant across layers *(Corollary 7)*.
>
> - Flat minima are spectral-norm balanced in depth-2 matrix factorization *(Corollary 8)*. This implies that **flat minima are not necessarily Frobenius-norm balanced, contrary to claims made in several works [1,2]**.
>
> - A minimizer of deep matrix factorization loss is flat **if** the product of spectral norms of left and right intermediate factors is constant across layers *(Corollary 10)*.
>
> We have added **a Discussion section, which was absent in the previous draft**. In this section, we discussed the further implications of the results presented in Section 5. We thoroughly examined claims in [1, 2] that flat minima coincide with Frobenius-norm balanced minima in depth-2 matrix factorization. We also discussed how misleading current sharpness measures can be for loss landscape analysis. We concluded this section by highlighting **the need for a new, robust sharpness measure that unifies the insights from existing ones.**
>
> Furthermore, we have revised the **entire Experiment section** to clarify issues related to $x _0$. Additionally, we moved the proof of Theorem 4 to the Appendix and revised Conclusion section.
>
> We noticed a parallel submission to ICLR [3] that empirically shows sharpness correlates most with the spectral norm of weight matrices in two-layer linear networks, compared to Frobenius and nuclear norms of weight matrices. They further demonstrate that, in deep linear networks, sharpness correlates most with the spectral norms of products of weight matrices, compared to the Frobenius and nuclear norms of products of weight matrices. With this regard, our work is the first to provide a theoretical justification for these empirical findings. We have incorporated this observation into the Section 1.1.
>
> [1] L. Ding, D. Drusvyatskiy, M. Fazel, and Z. Harchaoui, “Flat minima generalize for low-rank matrix recovery,” 2024.
>
> [2] "Learning Dynamics of Deep Linear Networks Beyond the Edge of Stability", Avrajit Ghosh, Soo Min Kwon, Rongrong Wang, Saiprasad Ravishankar, Qing Qu. ICLR 2025.
>
> [3] Anonymous, “Cracking the hessian: Closed-form hessian spectra for fundamental neural networks,” in Submitted to The Fourteenth International Conference on Learning Representations, under review, 2025.

---

### Author Response · Authors · 2025-11-15
**Main Revisions**

Dear reviewers,

First of all, we would like to thank you for your valuable insights, and we apologize for submitting the supplementary material and the main manuscript separately, in case this caused any inconvenience during the review process. We would like to remark the main revisions in the manuscript as follows:

**Empirical Evidence: A Parallel ICLR Submission.** We noticed a parallel submission to ICLR [1] that empirically shows sharpness correlates most with the spectral norm of weight matrices in two-layer linear networks, compared to Frobenius and nuclear norms of weight matrices. They further demonstrate that, in deep linear networks, sharpness correlates most with the spectral norms of products of weight matrices, compared to the Frobenius and nuclear norms of products of weight matrices. **With this regard, our work is the first to provide **a theoretical justification** for these empirical findings.** We have incorporated this observation into the Section 1.1.

**Local Minima.**  It is well known in the literature that overparameterized deep matrix factorization problems with convex and differentiable loss functions **do not have spurious minima**; in other words, every local minimum is global. In particular, this was shown in [2]. We updated the manuscript.


[1] Anonymous, “Cracking the hessian: Closed-form hessian spectra for fundamental neural networks,” in Submitted to The Fourteenth International Conference on Learning Representations, under review, 2025.

[2] T. Laurent and J. Brecht, “Deep linear networks with arbitrary loss: All local minima are global,” in ICML, PMLR, 2018.

---

### Author Response · Authors · 2025-11-28
**Major Revision!**

First, we would like to thank all reviewers for their valuable insights. In this draft, we have substantially revised the paper by addressing the main issues in the previous version. The revisions are as follows:

We added Section 5, **a new section that reveals remarkable aspects of flat minima** in general deep matrix factorization problems. These findings, which are **direct consequences of our Theorem 5**, are as follows:

- A minimizer of deep overparameterized scalar factorization loss is flat **if and only if** the product of spectral norms of left and right intermediate factors is constant across layers *(Corollary 7)*.

- Flat minima are spectral-norm balanced in depth-2 matrix factorization *(Corollary 8)*. This implies that **flat minima are not necessarily Frobenius-norm balanced, contrary to claims made in several works [1,2]**.

- A minimizer of deep matrix factorization loss is flat **if** the product of spectral norms of left and right intermediate factors is constant across layers *(Corollary 10)*.

We have added **a Discussion section, which was absent in the previous draft**. In this section, we discussed the further implications of the results presented in Section 5. We thoroughly examined claims in [1, 2] that flat minima coincide with Frobenius-norm balanced minima in depth-2 matrix factorization. We also discussed how misleading current sharpness measures can be for loss landscape analysis. We concluded this section by highlighting **the need for a new, robust sharpness measure that unifies the insights from existing ones.**

Furthermore, we have revised the **entire Experiment section** to clarify issues related to $x _0$. Additionally, we moved the proof of Theorem 4 to the Appendix and revised Conclusion section.

[1] L. Ding, D. Drusvyatskiy, M. Fazel, and Z. Harchaoui, “Flat minima generalize for low-rank matrix recovery,” 2024.

[2] "Learning Dynamics of Deep Linear Networks Beyond the Edge of Stability", Avrajit Ghosh, Soo Min Kwon, Rongrong Wang, Saiprasad Ravishankar, Qing Qu. ICLR 2025.

---

### Meta-Review · Area_Chair_uNxA · 2026-01-07

**Summary:**

This work provides an exact expression for computing the largest eigenvalue of the Hessian at minima in deep matrix factorization.

The reviewers gave consistent reviews that the derived expression is novel and regarded as the potentially important direction, but that only obtaining the expression does not reach the acceptance level of ICLR. The authors added further implications of their results in the rebuttal phase, but I think the reviewers would not be convinced without deeper explanations of the meaning of the result for the analysis of ML problems such as generalization, implicit regularization, and more non-trivial transient dynamics around minima, as suggested by the reviewers.

Thus, I evaluate this work as a rejection.

**Reviewer Concerns:**

In particular, the following concerns remain outstanding:

**Reviewer z1gb**
-  limited to the well-specified case (where the minimum loss is zero). Is it possible to extend the analysis to more general cases
-  further work may be needed ... such as analyzing the full Hessian structure or extending the results to local minima

**Reviewer BgPX**
- empirical claim ... has already been demonstrated (i.e., excape phenomena have been observed in the literature including Ghosh et al. and  in numerous works published prior to it)
- use the result from Theorem 5 to  theoretically demonstrate why unstable minima cannot sustain oscillations, which would strengthen the empirical story

**Reviewer F5xk**
-  having an explicit formula ... is a useful tool for further studies of generalization and implicit regularization. What exactly do we learn from having this representation? How can we use it?

**Reviewer Scores:**

The reviewers provided consistent reviews on the rejection side in a fairly confident manner, and I expect no score changes.

---

### Decision · Program_Chairs · 2026-01-26

Reject